# FORGE: Forming Semantic Identifiers for Generative Retrieval in Industrial Datasets

## Abstract

Semantic identifiers (SIDs) have gained increasing attention in generative retrieval (GR) due to their meaningful semantic discriminability. However, current research on SIDs faces three main challenges: (1) the absence of large-scale public datasets with multimodal features, (2) limited investigation into the generation strategies for better SIDs, whose assessment typically relies on costly GR training, and (3) slow online convergence in industrial deployment. To address these challenges, we propose **FORGE**, a comprehensive benchmark for **FO**rming semantic identifie**R**s for **G**enerative r**E**trieval in industrial datasets. Specifically, FORGE is equipped with a dataset comprising **14 billion** user interactions and multimodal features of **250 million** items sampled from one of the biggest e-commerce platforms in China, which serves over 300 million users each day. Leveraging this dataset, FORGE examines the impacts of SID construction on recommendations from multiple perspectives and validates their influence via offline experiments across different settings and tasks. Further online studies conducted on our platform for homepage recommendations show a 0.35% increase in transaction count, highlighting its practical impact. Regarding the expensive SID validation accompanied by full training of GRs, we propose two novel metrics of SID that correlate positively with the recommendation performance, enabling convenient evaluations without any GR training. For real-world applications, FORGE introduces an offline pretraining schema that reduces online convergence by half of the original. The code and data are available at `https://anonymous.4open.science/r/forge`.

## 1 Introduction

With the ability to predict the next item in an end-to-end manner, generative retrieval (GR) has recently emerged as a promising approach in recommender systems (Rajput et al., 2023; Liang et al., 2025). Typically, this framework encodes user behaviors into a sequence of *identifiers*, employs larger models to capture item dependencies, and generates *identifiers* of candidates as results. Therefore, the tokenization, which determines identifiers for each item, plays a fundamental role in advancing accurate recommendation (Lin et al., 2025).

Recent advancements in item tokenization for recommendation can be broadly categorized into single-token identifiers and semantic identifiers (SIDs). **i)** In Figure 1(a), the former randomly assigns a unique identifier to each item (e.g., 1892392→🐸) and updates the corresponding latent token independently during training. This schema would lead to the management of massive vocabularies with billions of items in real-world recommendations (Barkan & Koenigstein, 2016). Numerous identifiers prevent it from including all items, but only several negative ones are sampled for loss calculation to reduce computational burdens (Zhai et al., 2024; Ma et al., 2024). Such an approximation, however, might result in inconsistency between offline training and online serving, where the latter re-

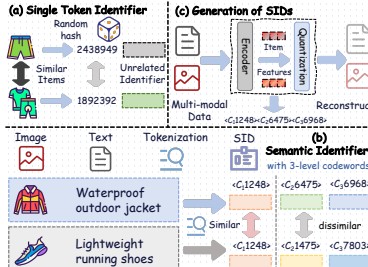

Figure 1: Diagram of (a) single token identifier and (b) semantic identifier. (c) The generation process of semantic identifiers.

quires measuring all items. **ii)** The semantic identifiers shown in Figure 1(b) represent each item with multi-level **codewords** (e.g., $<C_1 1248><C_2 6475><C_3 6968> \to$ 🎒 ), and recommenders would generate the top-K item identifiers as candidates with continuous beam search (Li et al., 2025; Zheng et al., 2025a). The construction of codewords within SID is based on multimodal data, thus enabling knowledge sharing among similar items. Furthermore, with hierarchical codewords, SIDs could compactly encode an exponentially large item space (e.g., a 3-level SID with 8192 tokens per level could represent $8192^3 \gg 10^{10}$ items using 24576 tokens). The relatively small number of tokens also makes it computationally feasible to include all of them in the calculations during training. (Rajput et al., 2023).

As depicted in Figure 1(c), the tokenization process for SIDs typically follows an encoder-quantization framework: an encoder for extracting semantic features from raw multimodal data and another quantization module (Esser et al., 2021; Lee et al., 2022) for SID encoding, separately (Zheng et al., 2024). These compact yet semantically rich identifiers subsequently remain unchanged and are used by downstream retrieval models during task-specific training. Despite their promising results on real-world datasets (Liu et al., 2025a; Ju et al., 2025), they still have the following challenges that hinder the development of this field:

1. **Limited behavior and features within datasets**. The majority of publicly available datasets in recommendation are limited in size ($\leq 10$ million) (Harper & Konstan, 2015; Ni et al., 2019), resulting in findings that may not generalize well when applied to industrial scenarios ($\geq 1$ billion) (Li et al., 2019). In addition, their construction is primarily based on single-token identifiers in the absence of abundant multimodal features, thus not accounting for SIDs (Zhu et al., 2021; 2022; Yuan et al., 2022). The dataset limitations impede the ability of later researchers to generate practical insights.

2. **Neglect of strategy selection for SID generation**. Even if SID serves as the foundation for the retrieval task (Yang et al., 2024; Ju et al., 2025), useful insights around the design choices for it are typically not well-discussed in previous literature, making it complicated for researchers to select an optimal configuration. Furthermore, recent works assess the quality of SIDs through time-consuming training of GR (Ju et al., 2025), remaining a critical need for direct metrics with convenient evaluations of SIDs.

3. **Inefficiency in industrial deployment**. Once a better SID configuration is confirmed via offline experiments, it typically requires deployment in the production environment and further training with the retrieval model for several days to reach convergence. The lengthy process hinders engineers from leveraging online observations to validate the SID optimizations adequately. Unfortunately, recent studies (Deng et al., 2025; Zheng et al., 2025b) consistently overlook the practical aspects for efficient deployment. This absence limits their broader impact and applicability in real industrial settings.

To mitigate this gap, we propose **FORGE** 🔨, a comprehensive benchmark for **FO**rming semantic identifie**R** in **G**enerative r**E**trieval with industrial datasets. (**Challenge 1**) In contrast to current public datasets that suffer from small scale and insufficient multimodal richness, FORGE is collected from one of the biggest e-commerce platforms in China, providing more than 14 billion user behaviors in 10 days, along with multimodal features of 250 million items. We partition the dataset into three consecutive phases to support continuous training and ensure robustly separated evaluation, facilitating convenient use by researchers. (**Challenge 2**) With the proposed dataset, FORGE investigates several strategies related to SID generation from multiple angles, encompassing diverse data modalities, SID encoding, and ID collision mitigation (i.e., cases where multiple similar items correspond to the same SID). We thoroughly validate the influence of these strategies across a range of settings, including different downstream GR models, SID structures, and search tasks. Following improvements observed in the 7-day online evaluation using the best SID strategies support the validity of our benchmark. In addition, FORGE introduces two novel metrics (i.e., embedding hitrate and Gini coefficient) for directly assessing SID quality. Experiments show that these metrics exhibit strong correlation with GR performance, enabling SID measurements without the need for costly GR training and thus providing an effective and efficient evaluation process. (**Challenge 3**) Serving as an industrial benchmark, FORGE also explores several warm-up techniques aimed at accelerating the convergence of new SIDs deployed in online scenarios. Continuous experiments spanning 10 days indicate that employing a

2-day offline pre-training allows the model to reach the same level as the base production recommenders in only half the time. We summarize the major contributions as follows:

1. To the best of our knowledge, FORGE releases the first dataset for generative retrieval with semantic identifiers built from industrial-scale user behaviors with rich multimodal features. Specifically, FORGE contains 100 times more user interactions than the largest existing recommendation dataset, along with 26 times more users and 3 times more items.

2. To enable researchers to identify which configuration benefits the SID generation, we explore several strategies from multiple perspectives, including data modality, SID encoding, and collision mitigation, to enhance the construction of SIDs. The offline experiments under different settings and tasks validate the reliability of our findings. Subsequent online improvement of 0.35% in transaction count further strengthens credibility.

3. For the SID evaluation, which requires expensive model training, we propose two novel metrics that positively correlate with GR performance, facilitating cost-free SID quality assessment without any GR training. Additionally, FORGE discovers an offline pretraining strategy that reduces the convergence time of new SIDs in production by half.

## 2 DATASET

Table 1: Dataset statistics and properties. FORGE contains a large-scale, temporally continuous dataset with multimodal content and a hierarchical codebook for semantic modeling.

| Dataset | Temporal Continuity | #Users | #Items | #Interactions | Modalities | Codebook | Tasks |
|---|---|---|---|---|---|---|---|
| MovieLens | N | 138K | 27K | 20M | ID, text | N | Recsys |
| Yelp | N | 2.1M | 160K | 8M | ID, text | N | Recsys |
| Taobao | N | 987K | 4M | 100M | ID | N | Recsys |
| TenRec | N | 5.0M | 3.7M | 142M | ID | N | Recsys |
| KuaiRec | N | 7K | 10K | 12M | ID, tags | N | Recsys |
| RecFlow | N | 42K | 82M | 38M | ID | N | Recsys |
| **FORGE** | **Y** | **131M** | **251M** | **14B** | **ID, text, image** | **Y** | **Recsys, Search** |

### 2.1 DESCRIPTION

FORGE pioneers the first industrial dataset in recommendation containing a set of pre-processed SIDs directly adopted for generative retrieval, as well as the multimodal features used to generate these identifiers. Collected from one of the largest e-commerce platforms in China, it is designed to bridge the gap between research and industry by enabling insights derived from it to be validated in industrial settings. Specifically, the dataset in FORGE contains two main types, where the examples are placed in the Appendix A.4:

**Seq Data** samples 40 million user data per day from real interactions over ten consecutive days for training. These interactions are partitioned into three sequential stages (S1, S2, and S3) based on their temporal order. For each training stage, a corresponding test set of 100,000 user sequences is sampled from the following day.

Each sequence is identified by the universally unique identifier of each page view (i.e., recommendation list presented to users). It mainly contains three fields: **i)** *action_seq* represents the sequence of historical interactions occurring before the current page view. To ensure efficient inference and compatibility with online deployment, we limit the maximum length of each sequence to 100 and truncate those longer sequences. **ii)** *target_item* refers to the items interacted with on the current page. For training, only the first clicked item is treated as the target. In contrast, all interacting items are considered as targets to compute metrics more comprehensively. **iii)** *query* is None for recommendation, while set to the corresponding query keywords for the search task, which are also collected from the real user inputs.

**Item Info**. Following previous benchmarks (Yuan et al., 2022; Liu et al., 2025b), each item in *Seq Data* is represented by a single-token identifier (item ID). To standardize the generation of SIDs, FORGE enriches each item ID with additional multimodal information, serving as input for the encoding processes. To preserve privacy, we release only the processed *multimodal embeddings* and do not include any raw data. Additionally, items $i^+$ *with*

*the most collaborative relations* of each item $i$ (i2i for simplicity) are provided as a field of each item and aid in SID generation in Section 3.1. For a better understanding of items, we employ named entity recognition to extract *keywords from item attributes*. We also release *constructed SIDs* in our open dataset, allowing better reproducibility by the community.

## 2.2 Dataset Usage

The benefits of the proposed dataset can be classified into three aspects. **i) Semantic identifiers generation**: with the provided multi-modal features in Item-Info, users could construct their SIDs with different configurations (e.g., different SID levels, SID postprocessing, SID Encoding, etc.) and subsequently assess the quality of SIDs using our proposed two new metrics in Section 3.4. **ii) Generative recommendation/search with semantic identifiers**: researchers could further evaluate the effectiveness of their constructed SID with both recommendation and search data we provided in Seq Data. Moreover, they could also perform a thorough comparison between their proposed models and other SOTA baselines using our datasets. **iii) Standard retrieval tasks using single token identifiers**: as we also release the original user interaction sequences in action_seq using single token identifiers, it would be convenient for users to perform and conduct experiments on traditional retrieval with our data.

## 2.3 Analysis and Comparison

As depicted in Table 1, FORGE distinguishes itself from existing recommendation benchmarks in two aspects. First, early datasets such as MovieLens-20M (Harper & Konstan, 2015) and Yelp (Asghar, 2016) are relatively small in scale, with limited user-item interactions, which makes them unsuitable for modeling large-scale traffic typically in industrial systems. Subsequent efforts (Gao et al., 2022; Liu et al., 2025b) alleviate this with abundant interactions from real-world platforms. In Table 1, FORGE further significantly extends the largest dataset, TenRec (Yuan et al., 2022), by approximately 100 times, providing over 14 billion user behaviors. Training and testing across multiple time periods also guarantee its reliability. Second, FORGE pioneers the SID construction in generative retrieval. Unlike previous datasets that typically include only one or two modalities, FORGE integrates three complementary modalities across billions of items to enable better SID generation. Moreover, the inclusion of query attributes allows it to be used for evaluating search tasks.

## 3 Overall Framework

This section provides an overview of generative retrieval in Figure 2. Specifically, it begins with SID generation (Section 3.1), followed by Section 3.2, which presents the postprocessing procedure to resolve SID collisions. In Section 3.3, we introduce the generative retrieval, along with the evaluation metrics for both SID and recommendation in Section 3.4.

## 3.1 SID Generation

**Capturing the underlying item semantics** through the multimodal content is a key step in generating SIDs. A natural approach is to leverage pretrained multi-modal large language models like CN-CLIP (Yang et al., 2022), which is made of $\mathcal{M}_{\text{text}}$ and $\mathcal{M}_{\text{image}}$ in Figure 2(a), to extract and fuse features $\mathcal{H}^i$ from text $\mathcal{I}^i_{\text{text}}$ and image $I^i_{\text{image}}$ of item $i$:

$$\mathcal{H}^i = \mathcal{M}_{\text{Fusion}}(\mathcal{H}^i_{\text{text}}, \mathcal{H}^i_{\text{image}}) = \text{Multimodal Fusion}(\mathcal{M}_{\text{text}}(\mathcal{I}^i_{text}), \mathcal{M}_{\text{image}}(\mathcal{I}^i_{image})). \quad (1)$$

The $\mathcal{M}_{\text{Fusion}}$ ultilizes QFormer (Jiang et al., 2024) to integrate information from two modalities. To further enhance the discriminability of these representations and reflect patterns from user behaviors, researchers could employ another contrastive loss $L_{\text{InfoNCE}}$. It encourages items that frequently co-occur to have more similar representations:

$$L_{\text{InfoNCE}} = f(\mathcal{H}^i_{\text{text}}, \mathcal{H}^{i^+}_{\text{text}}, \mathcal{H}^{i^-}_{\text{text}}) + f(\mathcal{H}^i_{\text{image}}, \mathcal{H}^{i^+}_{\text{image}}, \mathcal{H}^{i^-}_{\text{image}}) + f(\mathcal{H}^i, \mathcal{H}^{i^+}, \mathcal{H}^{i^-}), \quad (2)$$

where $f$ denotes the InfoNCE loss (Radford et al., 2021). The positive $i^+$ and negative $i^-$ samples are collected from i2i in *Item Info* and in-batch negative sampling, respectively.

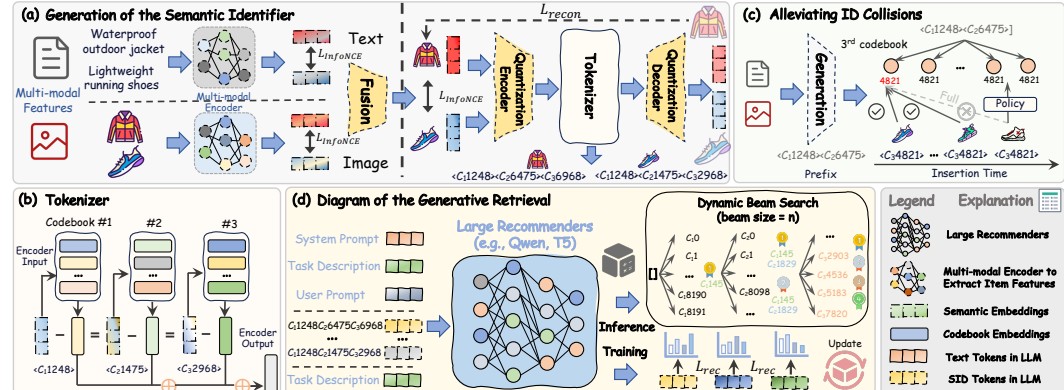

Figure 2: Overview of GRs. (a) The generation process of SIDs. Left panel: the training of the text/image encoder 🎨. Right panel: the schematic diagram of item tokenization. (b) Detailed workflow of the tokenizer. (c) Post-collision handling for semantic identifiers. (d) The training and dynamic inference procedure of generative retrieval.

**Item Tokenization.** The multi-modal feature $\mathcal{H}^i$ of item $i$ is then used to tokenize its SID. To ensure compatibility with the supervised learning framework and the autoregressive nature of generative retrieval, we select the RQ-VAE (Lee et al., 2022) in Figure 2(b) to convert $\mathcal{H}^i$ into a sequence of discrete codewords $\{c_1, ..., c_m\}$, with $m$ as the number of codewords. Moreover, in Appendix A.7, we also compare the RQ-VAE with alternatives like random assignment, multiple VQ (Esser et al., 2021), Optimized Product Quantization (OPQ) (Hou et al., 2025), and RQ-Kmeans (Deng et al., 2025) to validate the effectiveness.

## 3.2 ALLEVIATING ID COLLISIONS

During the SID generation, multiple similar items from different configurations or shops may be assigned the same SID, a situation referred to as **ID collision**. This may also result in low utilization, where certain SIDs remain unassociated with any item. To alleviate this issue, in Figure 2(c), we introduce two optional post-processing strategies to control the maximum number of items mapped to each SID. The **KNN-based** strategy samples multiple candidate SIDs and evaluates them sequentially based on their associated scores until a SID with fewer than $\sigma$ corresponding items is found, where $\sigma$ is a predefined threshold. In contrast, the **random-based** strategy focuses on dispersing the SID distribution at the final level rather than preserving semantic consistency. It assigns incremental codebooks to the last level in a circular manner (e.g., $\mathbf{C_m 0} \rightarrow C_m 1 \rightarrow ... \rightarrow C_m n_m - 1$ (the last codeword) $\rightarrow \mathbf{C_m 0}$), following the insertion order of items sharing the same prefix. Both approaches lead to a fairer distribution of codebook usage and further improve the performance of GR.

## 3.3 GENERATIVE RETRIEVAL

Once the set of SIDs is constructed, we concatenate the SIDs derived from user behavior with system instructions and user-specific prompts to form the input $x$ for the large recommender, which can be implemented with arbitrary architectures like Qwen (Team, 2024) or T5 (Ni et al., 2022). Tokens of codewords are extended into the tokenizer $\tau$ and jointly trained with the recommender $\theta$. Formally, given a sequence of codewords $\{c_1, c_2, ..., c_m\}$ representing the SID of the target item, we use cross-entropy loss (Mannor et al., 2005) to optimize both $\theta$ and $\tau$ by maximizing the conditional probability distribution $P_{\theta,\tau}(c_i|x, c_{j<i})$:

$$\mathcal{L}_{\text{rec}} = \sum_{i=1}^{m} log P_{\theta,\tau}(c_i|x, c_{j<i}). \tag{3}$$

For inference, we employ beam search (Freitag & Al-Onaizan, 2017) and retain the top-K SIDs iteratively as the retrieval results. However, maintaining a fixed beam width across all decoding steps can introduce significant computational overhead, particularly in real-time scenarios involving almost ten thousand requests per second. Under these circumstances,

each step must receive K sequences as input, resulting in a computational cost of K times. To address this, as illustrated in Figure 2(d), FORGE is equipped with **dynamic beam search**. Specifically, we set the beam size to 300 for the first, 600 for the second, and $K = 1200$ for the final decoding step. Therefore, the first decoding step retains only 300 sequences, and the second decoding step processes significantly fewer sequences compared to the non-dynamic approach, which would involve 1200 sequences as input. As a result, the computational cost of the second decoding step is reduced by 75% and the number of SIDs remains 1200 after the third decoding step, thereby balancing efficiency and effectiveness.

### 3.4 Evaluation Metrics

In this paper, we adopt the recommendation performance metric **HitRate** (HR) at the **item level** to evaluate the quality of generated SIDs. Taking a 3-level SID with 8192 codebooks per level as an example, **i**) FORGE first generates a list of 1200 candidate SIDs based on the beam search of GR. **ii**) We map these SIDs back to their corresponding original items. **iii**) The HitRate@K is measured by selecting the top-K items with the highest scores and checking whether each of them matches the ground truth items in the session. However, evaluation with HR requires full model training, which is typically resource-intensive and time-consuming. To address this, FORGE introduces **embedding hitrate** and **Gini coefficient** to assess the SID quality directly. **i)** Specifically, embedding hitrate regards the multi-modal features $\mathcal{H}^i$ as the semantic alternatives of traditional item embeddings. It is calculated with a separate item-to-item retrieval of all historical data based on the similarity (e.g., inner product) among the multi-modal features $\mathcal{H}^i$, providing a preliminary evaluation of $\mathcal{H}^i$. **ii)** The Gini coefficient measures the fairness in how identifiers are assigned across the item space. Given all of the SIDs $\{C_10C_20C_30,..., C_18191C_28191C_38191\}$ and the corresponding number of items assigned to each SID $I_c = \{5, 1, ..., 4\} \in \mathbb{N}^{N_d}$, we rank the SIDs in increasing order of $I_c$ to obtain a sorted list $S_{id} = \{SID_1, ..., SID_{N_d}\}$, where $N_d = 8192^3$ is the total number of SIDs. The calculation for the Gini coefficient is defined as:

$$G = \frac{2}{N_d} \sum_{i=1}^{N_d} (\frac{i}{N_d} - L(i)), \quad (4)$$

$$L(i) = \frac{C(i)}{C(N_d)}, \quad (5)$$

$$C(i) = \sum_{1}^{i} I_c(S_{id}^i). \quad (6)$$

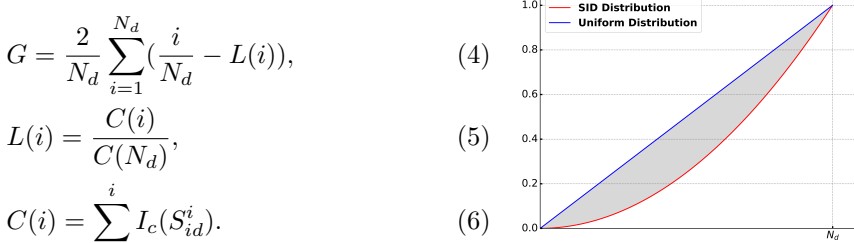

Figure 3: Intuitive explanation of the Gini Coefficient.

$I_c(S_{id}^i)$ is the associated item count of i-th SID in $S_{id}$. This can be regarded as the difference of the cumulative distribution function (CDF) between the uniform and the real SID distribution as exemplified in Figure 3. Our experiments demonstrate that both metrics exhibit a strong correlation with HR, enabling an efficient SID evaluation without the need for additional GR training. Detailed descriptions of other metrics are provided in Appendix A.6.

## 4 Experiments

In this section, we conduct experiments on our proposed datasets across three consecutive stages (S1, S2, and S3). For validation, we sample data from the day following each stage. By default, we employ RQ-VAE for SID generation and Qwen2.5-0.5B for generative retrieval. The goal of these experiments is to address the following research questions:

- **RQ1:** What generation strategies of SIDs are beneficial for downstream generative retrieval in terms of the hitrate?

- **RQ2:** In addition to the performance of GR accessed after time-consuming iteration, what evaluation criteria can be used to judge the quality of the SIDs more conveniently?

- **RQ3:** How is the generalization of SID optimization strategies proposed in this paper?

- **RQ4:** In practice, once a better SID configuration is identified offline in applications, how can it be deployed online to achieve fast convergence and obtain an effective observation?

Table 2: The effectiveness of SIDs optimizations compared with `base` where **bold** denotes the best method. `Base` is constructed using limited item-to-item collaborative information as defined in Equation 2, along with a KNN-based post-processing where the threshold $\sigma$ is set to 25, while `Ours` integrates `sideinfo`, `i2i` and `Random-5` together for SID generation.

| Stage | Method | HR@20 | | HR@100 | | HR@500 | | HR@1000 | |
|---|---|---|---|---|---|---|---|---|---|
| S1 | base | 3.61% | +0.00% | 9.67% | +0.00% | 20.82% | +0.00% | 25.09% | +0.00% |
| | +KNN-10 | 3.91% | +8.31% | 10.32% | +6.72% | 21.85% | +4.95% | 25.38% | +1.16% |
| | +KNN-5 | 4.18% | +15.79% | 10.81% | +11.79% | 22.14% | +6.34% | 25.30% | +0.84% |
| | +Random-5 | 4.67% | +29.36% | 11.79% | +21.92% | 23.65% | +13.59% | 26.30% | +4.82% |
| | +i2i | 3.79% | +4.99% | 10.04% | +3.83% | 21.71% | +4.27% | 26.97% | +7.49% |
| | +sideinfo | 3.81% | +5.54% | 10.00% | +3.41% | 21.69% | +4.18% | 26.88% | +7.13% |
| | Ours | **4.89%** | +35.46% | **12.31%** | +27.30% | **24.79%** | +19.07% | **29.01%** | +15.62% |
| S2 | base | 4.15% | +0.00% | 10.70% | +0.00% | 22.71% | +0.00% | 26.70% | +0.00% |
| | +KNN-10 | 4.63% | +11.57% | 11.52% | +7.66% | 23.24% | +2.33% | 26.80% | +0.37% |
| | +KNN-5 | 4.81% | +15.90% | 11.90% | +11.21% | 24.18% | +6.47% | 27.09% | +1.46% |
| | +Random-5 | 5.23% | +26.02% | 12.88% | +20.37% | 25.05% | +10.30% | 27.74% | +3.90% |
| | +i2i | 4.12% | -0.72% | 10.69% | -0.09% | 23.06% | +1.54% | 28.10% | +5.24% |
| | +sideinfo | 4.32% | +4.10% | 11.06% | +3.36% | 23.46% | +3.30% | 28.43% | +6.48% |
| | Ours | **5.33%** | +28.43% | **13.23%** | +23.64% | **26.37%** | +16.12% | **30.28%** | +13.41% |
| S3 | base | 4.33% | +0.00% | 11.16% | +0.00% | 23.26% | +0.00% | 27.24% | +0.00% |
| | +KNN-10 | 4.56% | +5.31% | 11.78% | +5.56% | 24.06% | +3.44% | 27.54% | +1.10% |
| | +KNN-5 | 5.08% | +17.32% | 12.61% | +12.99% | 24.66% | +6.02% | 27.72% | +1.76% |
| | +Random-5 | 5.43% | +25.40% | 13.14% | +17.74% | 25.39% | +9.16% | 28.03% | +2.90% |
| | +i2i | 4.22% | -2.54% | 11.32% | +1.43% | 23.69% | +1.85% | 28.91% | +6.13% |
| | +sideinfo | 4.34% | +0.23% | 11.43% | +2.42% | 24.20% | +4.04% | 29.32% | +7.64% |
| | Ours | **5.44%** | +25.64% | **13.79%** | +23.57% | **26.71%** | +14.83% | **30.86%** | +13.29% |

## 4.1 Overall Analysis about the Optimization of SIDs (RQ1)

Table 2 presents the ablation results based on the 3-level codewords with 8192 codebooks per level (i.e., 3×8192). From the results, we can conclude the following findings from two perspectives: **i) Post-processing to reduce collisions after SID generation leads to a more fair SID distribution and significantly improves retrieval performance.** `+KNN-5` and `+KNN-10` impose a limit on the number of items each SID can represent using the KNN-based solution. Relative better metrics from `base` to `+KNN-10` and `+KNN-5` confirm the benefit of increasing fairness and representation granularity. In contrast, `+Random-5` randomly assigns only the last codeword of each SID to an item, irrespective of semantic similarity. This method yields a more balanced SID distribution and further boosts retrieval performance, suggesting that the final layer of SIDs is less critical than maintaining high utilization or fairness across the SID space. **ii) Incorporating higher-quality modalities enhances the precision of SID construction.** `+i2i` introduces co-occurrence-based item relationships into the contrastive learning of Equation 2, while `+sideinfo` enriches the SID generation with additional textual item metadata (e.g., categories, sellers) by **concatenating the embeddings with the original text embedding**. Although `+i2i` slightly underperforms the `base` model in terms of HR@20, both strategies outperform the baseline at larger thresholds (e.g., HR@1000), achieving over 5% improvements across all training stages. This highlights the value of richer input signals in enhancing the effectiveness of GR. By integrating all these strategies, `ours` using more `sideinfo` and `i2i` data with `Random-5` for SID generation demonstrates consistently more than 13% increase compared to `base`.

## 4.2 Direct Metrics for the Evaluation of SIDs (RQ2)

The direct evaluations of SIDs using embedding hitrate and Gini coefficient are presented in Figure 4. As we mentioned in Section 3.4, the embedding hitrate captures the quality of

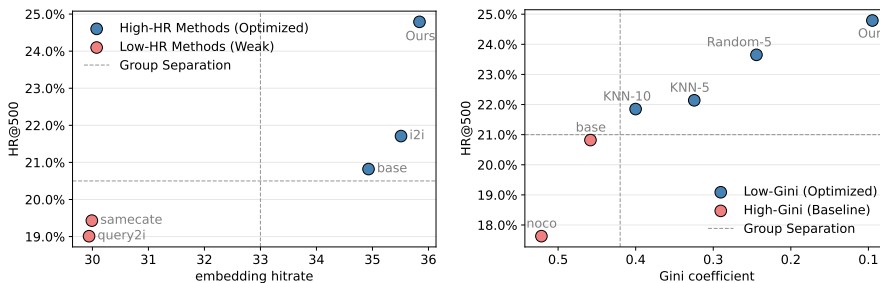

Figure 4: The effectiveness of embedding hitrate and Gini coefficient for SID evaluation. Left: Positive correlation between embedding hitrate and retrieval performance. Right: Lower Gini coefficient (fairer SID usage) leads to higher HR@500.

item collaborative relationships within the multi-modal feature $\mathcal{H}^i$, while the Gini coefficient reflects the fairness of SID distribution. **First**, it is evident that leveraging more related information could lead to the improvement of the associated metrics. From the figure 4, we could observe that incorporating richer relational information consistently improves these metrics. For instance, `i2i` with more collaborative item-item interactions achieves a higher embedding hitrate compared to the baseline `base` and methods like `query2i`. Likewise, a more refined SID collision strategy (e.g., `noco`→`base`→`KNN-10`→`KNN-5`→`Random-5`) could lead to higher codebook fairness and thus a lower Gini coefficient. We **then** train the retrieval model with all these SIDs across three stages, and surprisingly discover that both metrics have a strong relation with the ultimate hitrate. This motivates us that **it is not strictly necessary to train recommenders** to assess the quality of SIDs. Instead, our proposed metrics provide a reliable and efficient proxy for evaluating SID effectiveness, offering a practical and scalable solution for future research and applications.

### 4.3 Analysis about the Generalization of SID Optimization (RQ3)

All previous experiments were conducted using 3-level SIDs and decoder-only architecture Qwen 2.5-0.5B. In this part, we investigate whether the proposed optimizations are robust across different configurations, architectures, and domains. Therefore, we conduct additional experiments that explore variations in these dimensions.

**Impact of the Level of Codebook.** The comparison between `base` and `Ours` using 2-level codewords (i.e., 32768 codebooks each level) is placed in Table 3. This setup maintains the total representational capacity of the original multi-level design but distributes it across fewer levels. The improvements from `base` to `Ours` verify that our optimization generalizes well to configurations with fewer levels. Across three stages, `ours` achieves more than a 10% increase in HR@10 and HR@1000 over `base`.

Table 3: Performance of `base` and `Ours` under $2 \times 32768$ level codewords.

| Stage | Version | HR@20 | HR@100 | HR@500 | HR@1000 |
|---|---|---|---|---|---|
| S1 | base | 3.37% | 9.09% | 20.33% | 26.94% |
|    | Ours | **4.57%** | **11.94%** | **24.67%** | **31.34%** |
| S2 | base | 3.85% | 10.03% | 21.93% | 28.96% |
|    | Ours | **5.08%** | **12.61%** | **26.17%** | **33.53%** |
| S3 | base | 3.91% | 10.49% | 22.65% | 30.00% |
|    | Ours | **5.11%** | **13.04%** | **26.88%** | **34.15%** |

Table 4: Performance of `Ours` under different architecture and model size in Stage S1.

| Codebook | t5-base (0.2B) | | | | Qwen2.5-3B | | | |
|---|---|---|---|---|---|---|---|---|
| | HR@20 | HR@100 | HR@500 | HR@1000 | HR@20 | HR@100 | HR@500 | HR@1000 |
| 3×8192 (base) | 1.29% | 3.62% | 8.91% | 11.98% | 4.49% | 11.71% | 24.40% | 28.94% |
| 3×8192 (Ours) | **2.36%** | **5.99%** | **12.98%** | **16.08%** | **5.60%** | **14.18%** | **27.58%** | **32.12%** |
| 2×32768 (base) | 1.84% | 5.14% | 11.98% | 16.45% | 3.72% | 10.19% | 22.64% | 29.75% |
| 2×32768 (Ours) | **2.46%** | **6.55%** | **14.30%** | **18.88%** | **4.96%** | **12.80%** | **27.13%** | **34.29%** |

**Analysis of the Architecture.** The left panel of Table 4 presents the performance of `t5-base` (Raffel et al., 2020) with an encoder-decoder architecture. While it differs from the decoder-only setup we used in previous experiments, our proposed optimizations still yield

substantial improvements across different architectural paradigms. The consistent performance gain over the `base` model for both types of SIDs highlights our broad applicability.

**Influence of the Model Size.** The ability of FORGE can also be extended to larger models with 3B parameters, shown in the right part of Table 4. Moreover, increasing the model size consistently enhances retrieval performance, leading to nearly 10% improvements across all settings compared to the `Qwen2.5-0.5B` baseline in Table 2 and 3. We attribute this to the greater knowledge retention capacity enabled by the expanded parameter count.

Table 5: Retrieval performance on the generative search task across stages. `Ours` consistently outperforms the `base` counterparts under both codebook configurations.

| Stage | Codebook | HR@20 | HR@100 | HR@500 | HR@1000 |
|---|---|---|---|---|---|
| S1 | 3×8192 (base) | 14.30% | 27.91% | 43.95% | 50.74% |
|  | 3×8192 (Ours) | **23.39%** | **37.05%** | **51.77%** | **56.89%** |
|  | 2×32768 (base) | 11.53% | 24.09% | 39.65% | 47.21% |
|  | 2×32768 (Ours) | **17.77%** | **31.56%** | **46.98%** | **53.91%** |
| S2 | 3×8192 (base) | 14.93% | 28.98% | 46.21% | 53.10% |
|  | 3×8192 (Ours) | **24.20%** | **38.76%** | **54.08%** | **59.36%** |
|  | 2×32768 (base) | 12.14% | 25.38% | 41.77% | 49.47% |
|  | 2×32768 (Ours) | **18.89%** | **32.98%** | **49.14%** | **56.10%** |
| S3 | 3×8192 (base) | 15.08% | 29.83% | 47.83% | 55.10% |
|  | 3×8192 (Ours) | **24.71%** | **40.06%** | **56.04%** | **61.46%** |
|  | 2×32768 (base) | 12.54% | 26.26% | 43.48% | 51.57% |
|  | 2×32768 (Ours) | **19.13%** | **34.19%** | **51.18%** | **58.34%** |

**Generalization to the Search Domain.** In industrial scenarios, mobile apps often involve multiple tasks such as search and recommendation. Building a separate SID for each task can be resource-intensive. In this section, we investigate whether SIDs trained on the recommendation task can generalize to the search task. The input for the search task presented in Table 11 is similar to the recommendation, except for the query keywords of each user. As shown in Table 5, our findings in the search domain align closely with those observed in recommendation, where the 3-layer SID consistently outperforms the 2-layer SID. Moreover, across all configurations, the proposed SID optimization strategy demonstrates clear superiority over the baseline. This phenomenon highlights the robustness of FORGE, underscoring its potential for enabling versatile SIDs in real-world deployment scenarios.

**Investigation of the Data Modality.** We conducted more ablation experiments on the data modalities used in the multimodal representation. As shown in Table 6, using only text and image for representation generation leads to a significant performance drop. In comparison, the i2i-based method is more effective for recommendation tasks, but it still underperforms compared to the fusion of multiple modalities, especially when it comes to HR@1000. Therefore, all three modalities we incorporate contribute meaningfully to the generation of SIDs.

Table 6: Ablation study about the influence of the data modality during SID generation.

| Stage | Data Modality | HR@20 | HR@100 | HR@500 | HR@1000 |
|---|---|---|---|---|---|
| S1 | text-only | 3.01% | 7.97% | 18.00% | 21.01% |
|  | image-only | 3.06% | 8.34% | 18.79% | 22.03% |
|  | i2i-only | 4.63% | 11.58% | 22.52% | 23.15% |
|  | Ours | **4.89%** | **12.31%** | **24.79%** | **29.01%** |

**Online Study.** All the previous offline experiments were conducted solely at the retrieval stage, with evaluation metrics computed directly on the retrieved results without considering the subsequent ranking phase. To enable a more realistic and comprehensive evaluation pipeline,
we have deployed the model on the homepage of our platform for recommendations, where

Table 7: Online improvement with our SID optimization from 1st Sep to 7th Sep.

| PVR | Hitrate | Transaction Count |
|---|---|---|
| **+8.93%** | **+10.02%** | **+0.35%** |

the downstream ranking stage could further refine the retrieval items before presenting them to users. During serving, when a user clicks on an item detail page, an asynchronous inference request is triggered. The inference writes retrieval results into the cache table in 200ms, ensuring no impact on the main recommendation service's performance.

The improvement through online experiments is presented in Table 7 to serve as strong evidence, where our optimization improves the PVR and HR by 8.93% and 10.02% compared to `base`. Moreover, replacing `base` with `ours` ultimately results in a 0.35% absolute increase in user transactions.

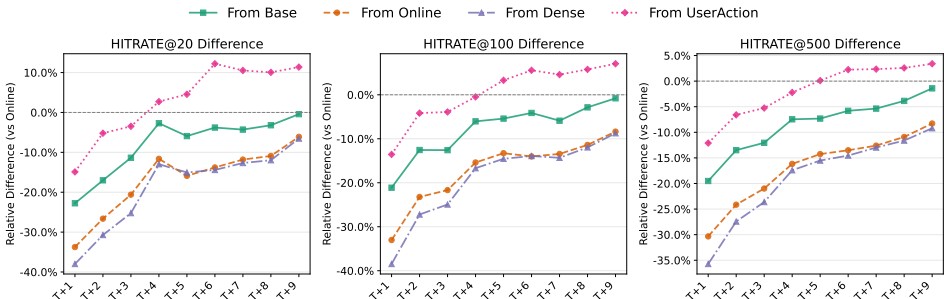

Figure 5: Relative difference from the production `base` version across cold-start days.

### 4.4 Practical Insights for Codebook Upgrade and Convergence (RQ4)

In contrast to offline training, models deployed in real-world production systems typically require training on daily-emerging behaviors ($\geq$ 200 million samples each day) for tens of days until convergence. Once we find a more powerful SID, thoroughly verifying it through comparison with production models requires much effort. To address this challenge, we propose several initialization strategies aimed at accelerating the convergence.

Specifically, `From Base` initializes the training using Qwen 2.5, while `From Online` starts from an existing production model equipped with SIDs of the same size (e.g., using the tokenizer and weights of the `base` as initialization for `Ours`). Following Qwen-VL (Bai et al., 2023), `From Dense` assumes that knowledge encoded in the production model can be beneficial for the new SID. It retains the weights from the production model but independently initializes the SID-related tokens. Finally, `From UserAction` performs 3-day offline pretraining using concatenated user behaviors from the previous 7 days, followed by streaming training. To ensure a fair comparison, all methods except `From UserAction` are additionally deployed online starting from days T-1 and T-2. Further details can be found in Appendix A.4.

Figure 5 illustrates the evolution of online performance across several days. The `From Base` strategy catches up with the production model after 10 days of training and requires even more time to outperform the baseline. Initializing from the production model does not consistently benefit the new SID, both `From Online` and `From Dense` underperform relative to `From Base`. With thorough pretraining, `From UserAction` surpasses the production model within four days and achieves a 10% improvement in Hitrate@10 after ten days of training. This demonstrates its potential to significantly accelerate deployment in practical settings.

## 5 Conclusion

In this paper, we introduce **FORGE**, the first industrial benchmark in generative retrieval containing 14 billion user interactions and multimodal features of 250 million items for semantic identifier construction. Leveraging this, we systematically explore significant yet less-explored configurations of SIDs and investigate several strategies from multiple angles. Extensive offline and online results validate the effectiveness of our configuration. Additionally, FORGE introduces two metrics for efficiently evaluating SID quality without requiring expensive GR training. As an industrial benchmark, we also present a practical pretraining schema that enables faster convergence in industrial settings. We hope that FORGE will serve as a valuable resource for the community, encouraging further innovation in this area.

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

## A  APPENDIX

### A.1  THE USE OF LARGE LANGUAGE MODELS (LLMs)

In this paper, we use publicly available LLMs as the backbone model of our experiments. As for writing, we use generative AI tools such as Grammarly solely to improve the clarity, coherence, and overall quality of the written text. All figures, preliminary drafts, and content are written independently by the authors without reliance on LLMs.

### A.2  RELATED WORK

**Generative Retrieval.**  The rapid development of recommender systems has drawn a surge of research attention in recent years (Bobadilla et al., 2013; Zhang et al., 2019; Yu et al., 2025). As the initial filtering component in the recommendation system, the retrieval stage quickly narrows down the item pool from billions to tens of thousands of candidates that users may be interested in. Traditional retrieval methods (Kang & McAuley, 2018; Hidasi et al., 2015; Li et al., 2019) encode users and candidate items into dense embeddings with a unified space, followed by Approximate Nearest Neighbors (ANN) to select several related items given each user. After KE (De Cao et al., 2021) introduces the notion of editing and retrieving via generation, generative retrieval (Rajput et al., 2023; Deng et al., 2025; Zheng et al., 2025b) becomes an innovative end-to-end approach for both recommendation and search, which directly decodes the identifiers of user-interested items in an auto-regressive way. Based on this, subsequent work adapted more suitable training frameworks (Liang et al., 2025; Liu et al., 2025a) and introduced strategies like reinforcement learning (Deng et al., 2025; Zheng et al., 2025b) to generate items that better align with user preferences.

**Semantic Identifier.**  The innovative SID widely adopted in both search (Sun et al., 2023; Zeng et al., 2024) and recommendation (Rajput et al., 2023) utilizes multiple codewords to build identifiers for items, enabling knowledge sharing among similar ones. Subsequent researchers (Zhang et al., 2025; Wu et al., 2025) investigated the generalization ability of generative retrieval when facing unseen new items. The widespread use in two domains has also encouraged some work (Penha et al., 2024; 2025; Zhao et al., 2025) that focuses on generalizing SIDs well in a joint setting. From the perspective of modality to build SIDs, several works (Wang et al., 2024b; Liu et al., 2025a; Wang et al., 2024a; Li et al., 2025) incorporate additional knowledge from user-item interactions to align the multimodal features of items, thereby equipping the generated SIDs with both semantic and collaborative information in recommendation. As for the encoding, OneRec (Deng et al., 2025) and RPG (Hou et al., 2025) tend to simplify the quantization process and accelerate the decoding process via RQ-Kmeans and optimized product quantization (OPQ) (Ge et al., 2013). In addition, FORGE focuses on the fairness and utilization of SID and codebooks. It provides more optimizations across all those perspectives, providing a comprehensive analysis of their influence with reliable empirical results. To the best of our knowledge, FORGE is the first industrial benchmark to analyze the significance of all these aspects in this community.

**Dataset Benchmarks.**  In recent years, several public datasets (Harper & Konstan, 2015; Asghar, 2016) have been introduced to support reproducible research by enabling fair comparisons under consistent settings. More recently, benchmarks such as Bars (Zhu et al., 2022) have aimed to unify and standardize these datasets, streamlining evaluation across retrieval and ranking tasks in recommendation systems. Despite their popularity and contributions, most existing datasets suffer from limited user behaviors. For example, widely used collections like Amazon (Ni et al., 2019) typically contain fewer than 10 million interactions, with average user interaction sequences shorter than 20. These limitations hinder their applicability to large-scale industrial platforms that process hundreds of millions of daily actions. To address this gap, FORGE, alongside other industrial benchmarks (Yuan et al., 2022; Liu et al., 2025b; Gao et al., 2022), introduces massive datasets derived from real-world systems. Compared to prior efforts, FORGE distinguishes itself through its larger scale, stronger focus on generative retrieval and SID generation, and the integration of rich multimodal features.

## A.3 Algorithm of the post-processing for ID collisions.

The pseudocodes of the KNN-based policy and the random-based policy are shown in Algorithm 1 and Algorithm 2, respectively.

---

**Algorithm 1** KNN-based Policy for alleviating ID Collisions

---

1: $\text{codeword1}, \text{codeword2}, \{\text{multiple\_codewords3}\} \leftarrow \text{QUANTIZATION}(\mathcal{H}^i)$
2: **for all** $\text{codeword3} \in \{\text{multiple\_codewords3}\}$ **do**
3:     $\text{SID} \leftarrow [\text{codeword1}][\text{codeword2}][\text{codeword3}]$
4:     **if** $\text{count\_items}(\text{SID}) < \sigma$ **then**              $\triangleright$ $\sigma$ is a predefined threshold
5:         **return** SID
6:     **end if**
7: **end for**

---

**Algorithm 2** Random-based Policy for alleviating ID Collisions

---

1: **Global** $\text{IndexMap} \leftarrow \{\}$         $\triangleright$ Dictionary: key=(c1,c2), value=index
2: $\text{codeword1}, \text{codeword2} \leftarrow \text{QUANTIZATION}(\mathcal{H}^i)$
3: $\text{key} \leftarrow (\text{codeword1}, \text{codeword2})$
4: **if** $\text{key} \notin \text{IndexMap}$ **then**
5:     $\text{IndexMap}[\text{key}] \leftarrow 0$         $\triangleright$ Initialize index to 0 on first occurrence
6: **end if**
7: $\text{codeword3} \leftarrow \text{IndexMap}[\text{key}]$
8: $\text{IndexMap}[\text{key}] \leftarrow (\text{IndexMap}[\text{key}] + 1) \bmod n_3$  $\triangleright$ $n_3$ is the size of third-level codebook
9: **return** $[\text{codeword1}][\text{codeword2}][\text{codeword3}]$

---

## A.4 Detailed Dataset Description

**Seq Data.** An example of our sequence data is presented in Table 8. Each interaction in the sequence is uniquely identified by the row number, representing a single page view from a user. The action_seq denotes the historical interactions used to generate the recommendation list for this page view. Items within each sequence are represented by item_ids and separated by commas. To enhance reliability, FORGE supports another dataset for the search task, whose data follows the structure of the recommendation, except for the inclusion of an extra query keyword. All behaviors are sampled from a real-world platform, with 40 million items sampled per day over ten days. The data is divided into three stages (S1, S2, S3), where S1 contains four days for a better warm-up, while S2 and S3 contain three days. An additional 100,000 samples are collected on the following day after each stage for evaluation. The observed performance improvements across these three stages could provide strong empirical evidence of the effectiveness of research.

**Item Info.** The mapping between item_ids in the action_seq and their corresponding semantic identifiers (SIDs) is detailed in the Item Info of Table 9, where each entry provides relevant metadata associated with a given item_id. In addition to the constructed SIDs of each item $i$ (`base` and `Ours`), the dataset includes its related item $i^+$ with the most co-occurrence relationship and the multimodal feature $\mathcal{H}^i$ derived by both the image and text of item $i$. To protect user privacy, FORGE does not make the actual title of items available. Instead, we employ named entity recognition (NER) to extract meaningful keywords that serve as anonymized substitutes, facilitating the understanding of collaborative items while safeguarding sensitive information.

**For Generative Retrieval Task.** The instructions for the generative retrieval and search tasks are presented in Table 10 and Table 11, respectively. Each user behavior sequence is encoded as continuous SIDs with no separator. The inputs consist of both system and user instructions, while the answer is only made up of three codewords of the target SID. Optionally, for the search task, the query keywords of each request are added to the user

Table 8: Example of our sequence data on recommendation and search tasks with segmented fields. For validation, all the interacted items during this page view are used as targets.

| task | data split | target_item | query | action_seq |
|---|---|---|---|---|
| Recsys | train | 835905354006 | / | 566476193770, 907995423965, 897259251809, 916138422508, 841619863136, 841665058110, 895390654577, 895382438483, 844457291108 |
| | test | 817445445744, 780409885810 | / | 546901080156, 626466181497, 804805887937, 819489082025, 919698707572, 680837256237, 611110770249, 696394590725, 738397879056 |
| Search | train | 762971121687 | 富勒烯精华液 | 937018558175, 936950943952, 947234504809, 666112772468, 928130569440, 938702048112, 940280943038, 918448888947, 792039516661 |
| | test | 562133894238, 567812038293, 610069547271 | 干艾叶 | 736070049413, 949041120166, 945601927485, 922276261406, 801633068800, 901162945401, 884983761715, 705197027339, 890974760840 |

Table 9: Example of Item Info.

| item_id | $\mathcal{H}^i \in R^{512}$ | semantic_id | related_item | title |
|---|---|---|---|---|
| 835905354006 | [0.12, 0.56, ..., 0.03] | [1203,2315,3576] | 787551011877 | 女装衬衫 |
| 855036080309 | [0.04, 0.02, ..., 0.05] | [2706,4659,2176] | 545092516562 | 大码睡衣男款 |

instruction and placed after the user sequence. The whole training process is to enable LLMs to understand the instructions and provide satisfying item lists given an input.

Table 10: Example of the instruction-tuning dataset format for next-item prediction in e-commerce recommendation. Each behavior sequence is represented as a series of semantic IDs (e.g., `C1220C8322C20452`). For a better LLM tokenization, for the $j$th-level codeword in our processed SID, we represent it with $c_j + \sum_{k=1}^{j-1} n_k$ (e.g., $C8193$ with $C_21$ as the 2nd codeword and 8192 codebooks each level). The original data is in Chinese, and an English translation is provided below for reader comprehension.

| Field | Content |
|---|---|
| system | 你是一个推荐系统，根据用户的历史行为，预测用户在电商场景的下一步行为。我会给你一串连续行为的语义编码，按照用户点击的时间顺序排列，每个行为用一个语义 ID 表示。 |
| user | 当前用户的历史行为如下：`C1220C8322C20452C6084C10195C20067C3256C14673C21112C7054C9412C18926C3021C10986C18869C3411C15297C23680C116C15928C19183C4405C11869C21320C7221C13888C16791C7743C16005C22091` 请预测用户在电商推荐场景后续行为的语义编码。 |
| answer | C3626C8758C22717 |

**Translations are provided solely for reader comprehension:**

- **system**: You are a recommendation system. Predict the user's next action in an e-commerce scenario based on their historical behavior. A sequence of semantic IDs will be provided, representing past interactions in chronological order, with each behavior encoded as one semantic ID.
- **user**: User's historical behavior sequence: `[long sequence]`. Please predict the semantic ID of the user's next behavior in the e-commerce recommendation scenario.
- **answer**: `C3626C8758C22717`

Table 11: Example of the instruction-tuning dataset format for next-item prediction in e-commerce search. Each behavior sequence is represented as a series of semantic IDs (e.g., `C1220C8322C20452`). The original data is in Chinese, and an English translation is provided below for reader comprehension.

| Field | Content |
|-------|---------|
| system | 你是一个搜索系统，根据用户的历史行为和当前用户搜索词，预测用户在电商场景的下一步行为。我会给你用户的搜索词和一串连续行为的语义编码，按照用户点击的时间顺序排列，每个行为用三个词表示。 |
| user | 当前用户的历史行为如下：C5299C12905C22291C1477C12826C20935C1931C9821C24180C853C14294C22645C2075C12389C16385C853C13340C20935C4894C14014C18136C6398C13220C22786C6398C12675C23264C4707C8594C18742C6398C14580C20512C6398C9530C19761 用户的搜索词为：短款小背心，请预测用户在电商搜索场景后续行为的语义编码 |
| answer | C1832C15680C24359 |

> **Translations are provided solely for reader comprehension:**
> - **system**: You are a search system. Predict the user's next action in an e-commerce scenario based on their historical behavior and current query. The behavior sequence is given as semantic IDs, each representing a triplet of attributes, in chronological order.
> - **user**: User's history: `[long sequence]`. Query: "short cropped vest". Predict the semantic ID of the next behavior.
> - **answer**: `C1832C15680C24359`

**For `From UserAction` Task.** As described in Section 4.4, the best-performed initialization is achieved through offline continue pretraining using interactions collected over the past 7 days. As the objective of the pretraining is to enable LLMs to swiftly adjust to evolved SIDs, we omit the instruction parts but only retain tokens in the answer. Specifically, all user behaviors during this period are organized sequentially based on the interaction time and optimized in an autoregressive way in Table 12. This offline pretraining allows LLMs to quickly grasp the structure and collaborative information of new SIDs, significantly accelerating convergence during online deployment.

Table 12: Example of the pretrained dataset in the UserAction task for coldstart during deployment. Each sequence is represented as a series of semantic IDs (e.g., `C1220C8322C20452`).

| Field | Content |
|-------|---------|
| corpus | C5299C12905C22291C1477C12826C20935C1931C9821C24180C853C14294C22645C2075C12389C16385C853C13340C20935C4894C14014C18136C6398C13220C22786C6398C12675C23264C4707C8594C18742C6398C14580C20512C6398C9530C19761C5727C15620C17409 |

## A.5 IMPLEMENTATION DETAILS

**Running environment.** The experiments are conducted on the PPU 810E, a GPU architecture developed by Alibaba. This platform achieves approximately 90% of the computational performance of the NVIDIA A100 GPU, while featuring a high-capacity memory system with 95.6 GB of on-board memory.

**Hyperparameter configurations.** Table 13 and 14 summarize the hyperparameters and configuration of models in the SID generation task and generative retrieval task, respectively. For the ID generation task, RQ-VAE employs symmetric 3-layer MLPs (512-64-512) with a 64D discrete codebook and CN-CLIP fusion. The training lasts for 150 epochs using

AdamW for stable convergence. For the generative retrieval task, we set the maximum lengths of the historical behaviors to 100, and employ two sequence-to-sequence models in our experiments: a `T5-Base` model and a `Qwen2.5-0.5B` model. Both models are fine-tuned on the same dataset with a maximum token length of 1280, where the maximum source and target lengths are set to 1024 and 256, respectively. The T5-Base model is trained for 4 epochs with a per-device batch size of 80, resulting in a total batch size of 1280 using 16 PPU-ZW810E GPUs. It uses a constant learning rate of $2 \times 10^{-4}$ and the AdamW optimizer with default parameters. In contrast, the Qwen2.5-0.5B-Instruct model is trained for a single epoch with a smaller per-device batch size of 40, also achieving the same total batch size of 1280 using 32 PPU-ZW810E GPUs. It adopts a linear learning rate scheduler with a base learning rate of $5 \times 10^{-5}$. Both models are trained using bfloat16 precision to improve training efficiency. The `Qwen2.5-3B` model follows the same training configuration as the `Qwen2.5-0.5B` model. All the SIDs are regarded as new tokens and included in the tokenizer of LLMs upon initialization. The dynamic beam size of 3-level and 2-level SIDs during inference is set to {300, 600, 1200} and {600, 1200}, respectively. All codes are freely available at `https://anonymous.4open.science/r/forge`.

Table 13: Core configuration of the RQ-VAE model and training setup.

| | Parameter | Value |
|---|---|---|
| Model | Model Type | RQ-VAE (RQVAE_EMBED_CLIP) |
| | Codebook Dim | 64 |
| | Input Dim | 512 |
| | Encoder Structure | 3-LayerMLP (512-256-256-64, ReLU) |
| | Decoder Structure | 3-LayerMLP (64-256-256-512, ReLU) |
| | Fusion Structure | Q-former |
| | Multimodal Encoder | CN-Clip(chinese-clip-vit-base-patch16) |
| Training | Epochs | 150 |
| | Warmup Epochs | 40 |
| | Learning Rate | 0.0002 |
| | Optimizer | AdamW (betas=[0.9, 0.999], eps=1e-8) |
| | LR Scheduler | cosine |
| | Batch Size | 2048 |

Table 14: Hyperparameters and training configurations of the two models used in the Generative Retrival Task.

| | T5-Base | Qwen2.5-0.5B-Instruct |
|---|---|---|
| Seq Length | 100 | 100 |
| Model Type | T5 | Qwen2.5 |
| Checkpoint | google-t5/t5-base | Qwen/Qwen2.5-0.5B-Instruct |
| Max Length | 1280 | 1280 |
| Max Source Length | 1024 | 1024 |
| Max Target Length | 256 | 256 |
| Epochs | 4 | 1 |
| Batch Size (per device) | 80 | 40 |
| Total Train Batch Size | 1280 | 1280 |
| Learning Rate | 2e-4 | 5e-5 |
| LR Scheduler | constant | linear |
| Optimizer | AdamW (betas=[0.9, 0.999], eps=1e-8) | |
| bf16 | ✓ | ✓ |
| GPU Count | 16 | 32 |
| GPU Type | PPU-ZW810E | PPU-ZW810E |
| Training Time | 20h/epoch | 80h/epoch |

A.6 Evaluation Metric

This section provides a detailed explanation and calculation of the metrics employed in our paper. We primarily utilize the HitRate to evaluate retrieval performance, while the Embedding HitRate and Gini Coefficient are adopted for the preliminary assessment of SID quality. Additionally, other metrics used for both online and more refined SID evaluation in Figure 7, are also described.

**HitRate.** It measures whether any of the interacted items during the current page view of each user appear in the retrieval results of recommenders. For the retrieval, all items are ranked according to the predicted scores, and we truncate the list to select the top-K items as the retrieval results. The HR@K can be formulated as

$$HR@K = \frac{1}{N} \sum_{m=1}^{N} \frac{I_K \ \& \ I_{\text{click}}}{I_{\text{click}}}, \tag{7}$$

where $N$ is the total number of test samples, $I_{\text{click}}$ represents the set of items actually clicked by the user, and $I_K$ denotes the set of top-K retrieved items.

**PVR.** Real-world recommender systems typically consist of the retrieval and ranking stages, where there exist multiple retrieval models for online serving. The ranking model accepts thousands of candidates from these retrieval models and performs more delicate scoring on them. Only dozens of items are ultimately exposed to users as a single page view. PVR computes the ratio of items displayed to users ($PV_i$) that are successfully retrieved by our retrieval model, which can be formulated as follows:

$$\text{PVR} = \frac{\sum_{mt(i)=FORGE} PV_i}{\sum_I PV_i}, \tag{8}$$

where $mt(i) = FORGE$ indicates that the exposed item $i$ is retrieved by FORGE.

**Transaction Count.** It is a direct metric that quantifies the number of items successfully purchased by users during a given page view. As such, it serves as one of the most critical indicators of online performance, capturing the real-world impact of the methods in industry. A higher transaction count indicates that the recommended items are not only relevant but also compelling enough to lead to actual purchases.

**Feature Fidelity.** It is specifically designed to preserve the original semantic information and avoid its complete loss in the encoding process of RQ-VAE. A higher feature fidelity indicates that the multimodal knowledge is better preserved during quantization.

$$\text{Feature Fidelity} = \max \left( 0, 1 - \frac{\|\mathcal{H}^i - \mathcal{D}(r_1^{c_1} + ... + r_m^{c_m})\|_2}{\|\mathcal{H}^i\|_2} \right) \times 100\%. \tag{9}$$

**Style/Homogeneous Consistency.** In e-commerce scenarios, there are often multiple items in different shops that are either identical (style) or originate from the same source (*homogeneous*, as exemplified in Figure 6). The former refers to two items that are totally identical in size, category, and other attributes. The latter describes two items that are derived from each other through data augmentation techniques such as image editing or text modification. These two metrics evaluate whether items identified as the same style or same origin are assigned to the same SID:

$$\text{Style/Homogeneous} = \frac{\text{Item pairs with the same style/origin in the same SID}}{\text{Total number of item pairs with the same style/origin}} \tag{10}$$

**Codebook Utilization.** While the Gini Coefficient offers a high-level view of how evenly items are distributed across semantic identifiers (SIDs), it does not directly reflect the actual usage of the available codebook space. We thus provide another metric, codebook utilization, which measures the proportion of codebooks that are actively used to represent items.

$$\text{Utilization} = \frac{|\text{Non-empty codebooks}|}{|\text{Total codebooks}|} \tag{11}$$

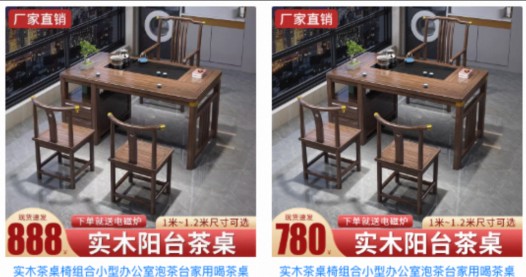

Figure 6: Example of item pairs with the same origin (Homogeneous).

A.7 ADDITIONAL EXPERIMENTAL RESULTS

In this section, we conduct more ablation studies on the different configurations of SID optimization to provide a clearer understanding of our methods for researchers.

**SID Collision.** Table 15 clarifies the impact of different ID collision strategies on the retrieval models. Compared to the `base` method, which employs a KNN-based strategy to limit the number of items per SID to fewer than 25, the `noco` variant without any post-processing shows significantly reduced effectiveness. Furthermore, results also indicate that more effective collision avoidance (`noco`→`base`→`KNN-10`→`KNN-5`) contributes to higher semantic encoding fairness and leads to higher metrics.

To further validate this relationship, we conduct an inverse experiment using the `merge` method, which intentionally degrades the SID fairness by merging adjacent SIDs below a certain threshold. The substantial drop in performance compared to `base` confirms a strong positive correlation between codebook utilization and retrieval accuracy.

Distinguished from the KNN-based policy, `Random-5` leverages a random-based approach to assign the last level codeword sequentially and randomly. While this reduces collisions and maintains high codebook utilization and SID fairness, it sacrifices meaningful semantic alignment at the final quantization level. The continuous hit rate gain suggests that utilization and collision are more important than the semantic features, as features in the last level capture limited information in the residual quantization.

**Data Modality for SID.** The i2i (collaborative items) relationships in recommendation systems can provide additional benefits for SID generation. Both the query2i and samecate (items within the same category) approaches underperform the `base` method, which utilizes only limited i2i data for alignment as in Equation 2. Besides, Table 16 demonstrates that incorporating item-side information (such as seller, price, and category) along with richer i2i relations could lead to more meaningful SIDs and improved retrieval performance.

**SID Level Structures.** In previous experiments, we primarily considered SID level structures such as 3×8192 or 2×32768. This naturally raises the question of whether alternative level structures may offer improved performance in generative retrieval tasks, such as those with more codewords (e.g., 4×4096) or distinct hierarchical setups (e.g., 2048_4096_8192). Extensive validation presented in Table 17 indicates that 3-level SIDs still represent the optimal trade-off between computational efficiency and retrieval effectiveness. In contrast, 2-level structures exhibit significantly worse performance, while 4-level structures incur higher inference costs with negligible gains. Notably, the 1024_4096_32768 configuration achieves more than a 10% improvement in HR@1000 compared to `base`.

**SID Quantization Methods.** As other related works (Deng et al., 2025; Zhou et al., 2025; Chen et al., 2025) have explored alternative encoding methods such as RQ-Kmeans, we also conduct a comparative analysis and present the results in Table 18. To ensure fairness and compatibility with our internal engineering infrastructure when comparing with OPQ and Multiple-VQ, we only use their encoding methods with KNN-25 for SID collisions, while relying on autoregressive generation combined with dynamic beam search for GRs. First, both the semantic-related and residual decoding strategies play a critical role

Table 15: Relative improvements of different ID collision alleviating strategies over the `base` model (3×8192). Improvements are computed as $(M - \text{base})/\text{base}$.

| Stage | Method | HR@20 | | HR@100 | | HR@500 | | HR@1000 | |
|---|---|---|---|---|---|---|---|---|---|
| S1 | base | 3.61% | +0.00% | 9.67% | +0.00% | 20.82% | +0.00% | 25.09% | +0.00% |
| | noco | 2.96% | -18.01% | 7.69% | -20.48% | 17.63% | -15.32% | 22.27% | -11.24% |
| | merge | 3.00% | -16.90% | 8.32% | -13.96% | 18.66% | -10.37% | 24.22% | -3.47% |
| | +KNN-10 | 3.91% | +8.31% | 10.32% | +6.72% | 21.85% | +4.95% | 25.38% | +1.16% |
| | +KNN-5 | 4.18% | +15.79% | 10.81% | +11.79% | 22.14% | +6.34% | 25.30% | +0.84% |
| | +Random-5 | 4.67% | +29.36% | 11.79% | +21.92% | 23.65% | +13.59% | 26.30% | +4.82% |
| S2 | base | 4.15% | +0.00% | 10.70% | +0.00% | 22.71% | +0.00% | 26.70% | +0.00% |
| | noco | 3.49% | -15.90% | 8.91% | -16.73% | 19.47% | -14.27% | 24.07% | -9.85% |
| | merge | 3.54% | -14.70% | 9.11% | -14.86% | 20.34% | -10.44% | 26.06% | -2.40% |
| | +KNN-10 | 4.63% | +11.57% | 11.52% | +7.66% | 23.24% | +2.33% | 26.80% | +0.37% |
| | +KNN-5 | 4.81% | +15.90% | 11.90% | +11.21% | 24.18% | +6.47% | 27.09% | +1.46% |
| | +Random-5 | 5.23% | +26.02% | 12.88% | +20.37% | 25.05% | +10.30% | 27.74% | +3.90% |
| S3 | base | 4.33% | +0.00% | 11.16% | +0.00% | 23.26% | +0.00% | 27.24% | +0.00% |
| | noco | 3.59% | -17.09% | 9.24% | -17.20% | 20.29% | -12.77% | 24.84% | -8.81% |
| | merge | 3.56% | -17.78% | 9.55% | -14.43% | 20.65% | -11.22% | 26.86% | -1.40% |
| | +KNN-10 | 4.56% | +5.31% | 11.78% | +5.56% | 24.06% | +3.44% | 27.54% | +1.10% |
| | +KNN-5 | 5.08% | +17.32% | 12.61% | +12.99% | 24.66% | +6.02% | 27.72% | +1.76% |
| | +Random-5 | 5.43% | +25.40% | 13.14% | +17.74% | 25.39% | +9.16% | 28.03% | +2.90% |

Table 16: Relative improvements of different data modalities over the `base` model (3×8192). Improvements are computed as $(M - \text{base})/\text{base}$.

| Stage | Method | HR@20 | | HR@100 | | HR@500 | | HR@1000 | |
|---|---|---|---|---|---|---|---|---|---|
| S1 | base | 3.61% | +0.00% | 9.67% | +0.00% | 20.82% | +0.00% | 25.09% | +0.00% |
| | query2i | 3.20% | -11.36% | 8.70% | -10.03% | 19.01% | -8.69% | 23.93% | -4.62% |
| | samecate | 3.44% | -4.71% | 8.92% | -7.76% | 19.43% | -6.68% | 23.94% | -4.58% |
| | +sideinfo | 3.81% | +5.54% | 10.00% | +3.41% | 21.69% | +4.18% | 26.88% | +7.13% |
| | +i2i | 3.79% | +4.99% | 10.04% | +3.83% | 21.71% | +4.27% | 26.97% | +7.49% |
| S2 | base | 4.15% | +0.00% | 10.70% | +0.00% | 22.71% | +0.00% | 26.70% | +0.00% |
| | query2i | 3.71% | -10.60% | 9.69% | -9.44% | 20.67% | -8.98% | 25.72% | -3.67% |
| | samecate | 3.81% | -8.19% | 9.86% | -7.85% | 21.31% | -6.16% | 25.78% | -3.45% |
| | +sideinfo | 4.32% | +4.10% | 11.06% | +3.36% | 23.46% | +3.30% | 28.43% | +6.48% |
| | +i2i | 4.12% | -0.72% | 10.69% | -0.09% | 23.06% | +1.54% | 28.10% | +5.24% |
| S3 | base | 4.33% | +0.00% | 11.16% | +0.00% | 23.26% | +0.00% | 27.24% | +0.00% |
| | query2i | 3.67% | -15.24% | 9.99% | -10.48% | 21.77% | -6.41% | 27.05% | -0.70% |
| | samecate | 3.98% | -8.08% | 10.23% | -8.33% | 21.91% | -5.80% | 26.50% | -2.72% |
| | +sideinfo | 4.34% | +0.23% | 11.43% | +2.42% | 24.20% | +4.04% | 29.32% | +7.64% |
| | +i2i | 4.22% | -2.54% | 11.32% | +1.43% | 23.69% | +1.85% | 28.91% | +6.13% |

in retrieval performance. The `Random` baseline assigns SIDs to items without considering any multimodal features, leading to a significant 70% drop in HR@1000. In contrast, the `Multiple-VQ` approach employs separate encoders at each level, resulting in decoupled codewords within SIDs and consequently suboptimal representation learning. Second, `RQ-Kmeans` and `OPQ` achieve performance comparable to `RQ-VAE` (base). As noted by OneRec (Deng et al., 2025), a key advantage of RQ-Kmeans lies in its efficient use of codebooks. However, through appropriate post-processing to handle ID collisions, `ours` optimized implementation with `RQ-VAE` demonstrates a clear superiority over `RQ-Kmeans`.

Table 17: Relative improvements of different SID level structures over the `base` model (3×8192). Improvements are computed as $(M - base)/base$.

| Stage | Method | HR@20 | | HR@100 | | HR@500 | | HR@1000 | |
|---|---|---|---|---|---|---|---|---|---|
| | base | 3.61% | +0.00% | 9.67% | +0.00% | 20.82% | +0.00% | 25.09% | +0.00% |
| | 2048_4096_8192 | 3.20% | -11.36% | 8.70% | -10.03% | 19.01% | -8.69% | 23.93% | -4.62% |
| | 8192_4096_2048 | 3.44% | -4.71% | 8.92% | -7.76% | 19.43% | -6.68% | 23.94% | -4.58% |
| S1 | 2x32768 | 3.37% | -6.65% | 9.09% | -6.00% | 20.33% | -2.35% | 26.94% | +7.37% |
| | 3x512 | 2.84% | -21.33% | 7.80% | -19.34% | 17.63% | -15.32% | 23.33% | -7.01% |
| | 1024_4096_32768 | 3.76% | +4.16% | 10.09% | +4.34% | 22.03% | +5.81% | 27.95% | +11.40% |
| | 4x4096 | 3.65% | +1.11% | 9.50% | -1.76% | 20.25% | -2.74% | 25.06% | -0.12% |
| | base | 4.15% | +0.00% | 10.70% | +0.00% | 22.71% | +0.00% | 26.70% | +0.00% |
| | 2048_4096_8192 | 3.96% | -4.58% | 10.44% | -2.43% | 22.17% | -2.38% | 27.64% | +3.52% |
| | 8192_4096_2048 | 4.02% | -3.13% | 10.52% | -1.68% | 22.54% | -0.75% | 28.21% | +5.66% |
| S2 | 2x32768 | 3.85% | -7.23% | 10.03% | -6.26% | 21.93% | -3.43% | 28.96% | +8.46% |
| | 3x512 | 3.34% | -19.52% | 8.65% | -19.16% | 19.46% | -14.31% | 24.89% | -6.78% |
| | 1024_4096_32768 | 4.34% | +4.58% | 11.07% | +3.46% | 23.58% | +3.83% | 29.75% | +11.42% |
| | 4x4096 | 4.19% | +0.96% | 10.53% | -1.59% | 21.49% | -5.37% | 26.11% | -2.21% |
| | base | 4.33% | +0.00% | 11.16% | +0.00% | 23.26% | +0.00% | 27.24% | +0.00% |
| | 2048_4096_8192 | 4.04% | -6.70% | 10.50% | -5.91% | 23.15% | -0.47% | 29.07% | +6.72% |
| | 8192_4096_2048 | 4.19% | -3.23% | 10.92% | -2.15% | 23.59% | +1.42% | 28.99% | +6.42% |
| S3 | 2x32768 | 3.91% | -9.70% | 10.49% | -6.00% | 22.65% | -2.62% | 30.00% | +10.13% |
| | 3x512 | 3.49% | -19.40% | 9.07% | -18.73% | 19.99% | -14.06% | 25.96% | -4.70% |
| | 1024_4096_32768 | 4.30% | -0.69% | 11.17% | +0.09% | 24.14% | +3.78% | 30.35% | +11.42% |
| | 4x4096 | 4.00% | -7.62% | 10.46% | -6.27% | 21.72% | -6.62% | 26.45% | -2.90% |

Table 18: Comparison of different quantization methods across stages.

| Stage | Method | HR@20 | HR@100 | HR@500 | HR@1000 |
|---|---|---|---|---|---|
| | Random | 1.08% | 2.91% | 4.82% | 4.82% |
| | Multiple-VQ | 2.87% | 8.15% | 18.30% | 23.84% |
| S1 | RQ-Kmeans | 3.42% | 9.39% | 20.71% | 26.11% |
| | OPQ | 3.57% | 9.57% | 21.19% | 26.64% |
| | RQ-VAE (base) | 3.61% | 9.67% | 20.82% | 25.09% |
| | RQ-VAE (Ours) | **4.89%** | **12.31%** | **24.79%** | **29.01%** |
| | Random | 1.66% | 4.09% | 6.66% | 6.66% |
| | Multiple-VQ | 3.33% | 8.68% | 19.55% | 25.36% |
| S2 | RQ-Kmeans | 3.79% | 10.09% | 22.23% | 27.95% |
| | RQ-VAE (base) | 4.15% | 10.70% | 22.71% | 26.70% |
| | RQ-VAE (Ours) | **5.33%** | **13.23%** | **26.37%** | **30.28%** |
| | Random | 1.51% | 4.04% | 6.41% | 6.41% |
| | VQ | 3.47% | 9.28% | 20.69% | 26.47% |
| S3 | RQ-Kmeans | 3.96% | 10.82% | 23.00% | 28.71% |
| | RQ-VAE (base) | 4.33% | 11.16% | 23.26% | 27.24% |
| | RQ-VAE (Ours) | **5.44%** | **13.79%** | **26.71%** | **30.86%** |

**Comparison with Vector Retrieval.** We compared the generative retrieval with those widely used vector retrieval approaches such as YouTubeDNN (Covington et al., 2016), SAS-Rec (Kang & McAuley, 2018), BERT4Rec (Sun et al., 2019), and HSTU (Zhai et al., 2024).

During evaluation, vector retrieval-based methods require computing the inner product with over 200 million item embeddings from S1, which significantly increases the inference cost. To facilitate reproducibility, we sampled and constructed a small dataset from the test samples of S1 to decrease the number of items. Due to the multi-level representation of SID, SID-based generative recommendation can represent all items with only a small number of tokens compared to traditional vector retrieval. This characteristic enables the model to calculate the loss function using all tokens during training without the need for negative sampling, thus avoiding the inconsistency between the sampled space during training and the full space when serving. The results in Table 20 depict that taking all the tokens into account when training enables more precise model updates over the entire sample space, where *ours* outperforms those vector retrieval-based ones. Equipped with SID, SAS-Rec_SID (5.34%) also shows a substantial improvement compared to the vanilla SASRec (3.12%), indicating that the semantic discriminability within SID may also have a positive influence on recommendations.

**Experiments on Public Datasets.** The results of FORGE on the Amazon2023-Clothing_Shoes_and_Jewelry (Hou et al., 2024), which consists of 7.2M items and 66.0M interactions, are shown in Table 19. We construct 3-level codewords with 512 codebooks per level (i.e., 3×512). The results show consistent improvements of Ours over base, demonstrating that our proposed methods remain robust on smaller-scale datasets.

Table 19: Generalization of FORGE on the Amazon Dataset.

| Method | HR@20 | HR@100 | HR@500 | HR@1000 |
|---|---|---|---|---|
| 3×512 (base) | 1.02% | 2.47% | 5.71% | 8.07% |
| 3×512 (Ours) | 2.12% | 4.56% | 9.79% | 13.04% |

Table 20: Comparison between generative retrieval with vector retrieval-based methods in Stage S1-Tiny.

| Stage | Method | HR@20 | HR@100 | HR@500 | HR@1000 | Params (Emb/Dense) |
|---|---|---|---|---|---|---|
| | YoutubeDNN | 0.07% | 0.43% | 1.74% | 2.26% | 1.573B/57.8k |
| | SASRec | 0.15% | 0.59% | 1.93% | 3.12% | 1.573B/61.9k |
| | Bert4Rec | 0.09% | 0.62% | 2.11% | 3.26% | 1.573B/61.9k |
| S1-Tiny | SASRec_SID | 0.23% | 1.24% | 3.65% | 5.34% | 1.574B/61.9k |
| | HSTU | 0.21% | 0.77% | 2.40% | 3.71% | 1.573B/224k |
| | Ours | **0.88%** | **2.32%** | **5.62%** | **7.07%** | 0.18B/0.35B |

**Additional Evaluation Metrics.** We further calculate extensive metrics described in Section A.6. **i)** As we proposed in previous content, the embedding hitrate, fairness, and utilization are examined to be positive with retrieval performance. The consistent ranking `Ours` > `base` > `RQ-Kmeans` > `Multiple-VQ` observed in both Figure 7 and Table 18 also validates this conclusion. **ii** Other metrics like feature fidelity do not show a clear correlation with retrieval effectiveness. `Multiple-VQ` achieves the best scores on two of them, but in Table 18 we can observe that it obtains the worst retrieval results. A similar pattern is observed for `RQ-Kmeans`, which significantly outperforms `base` in feature fidelity but still lags in overall retrieval performance.

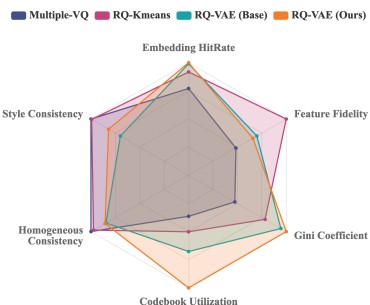

Figure 7: Performance of RQ-VAE under different metrics compared with other methods.

A.8 Optimized lm_head

---

**Algorithm 3** Efficient Logits Computation during Training

---

1: **function** FORWARD($input\_ids, labels$)
2:     $outputs \leftarrow$ BASE\_MODEL($input\_ids$)
3:     $hidden\_states \leftarrow outputs.last\_hidden\_state$
4:     **if** training **then**
5:         $first\_non\_neg \leftarrow$ MIN\_FIRST\_NON\_NEG\_INDEX($labels$)
6:         $logits\_to\_keep \leftarrow labels.size(1) - first\_non\_neg + 1$
7:         $labels \leftarrow labels[:, -logits\_to\_keep :]$
8:     **end if**
9:     $slice\_indices \leftarrow$ SLICE($-logits\_to\_keep, None$)
10:     $logits \leftarrow$ LM\_HEAD($hidden\_states[:, slice\_indices, :]$)
11:     $loss \leftarrow$ CROSS\_ENTROPY($logits, labels$)
12:     **return** loss, logits, outputs
13: **end function**
14: **function** MIN\_FIRST\_NON\_NEG\_INDEX($labels$)
15:     $mask \leftarrow (labels \geq 0).cumsum(dim = -1)$
16:     $first\_non\_neg \leftarrow$ ARGMAX($(mask == 1).float(), dim = -1$)
17:     **return** $min(first\_non\_neg)$
18: **end function**

---

To reduce the computational overhead during training, we only compute the necessary logits starting from the first valid label position in each sequence. This is achieved in Algorithm 3 by identifying the first non-negative label index using the `min_first_non_neg_index` function, and slicing the hidden states accordingly before applying the language model head. Figure 8 clearly shows that the optimization significantly reduces both memory usage and computation time.

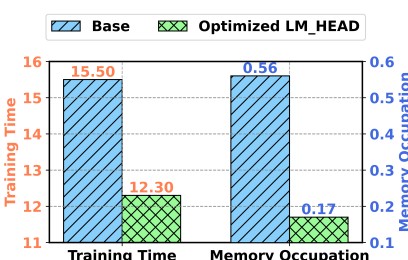

Figure 8: Efficiency gain with optimized LM\_HEAD.

### A.9 CASE STUDY

**SID Distribution.** Due to the large number of the SIDs of the 2nd level ($8192^2$), we only applied t-SNE (Wu et al., 2025) projection on the first-level codebooks trained using different quantization methods. Figure 9 illustrates their clustering patterns in a 2D space, where darker points indicate that an SID corresponds to more items. First, all points are distributed across the entire region, suggesting that they are semantically distinct from each other. Besides, compared to VQ-based methods, the RQ-VAE without any post-processing for SID conflicts can achieve a lower collision rate, as evidenced by significantly fewer points with items greater than 1000. Additionally, the points in the right panel are more dispersed, indicating that the codebooks exhibit stronger discriminability.

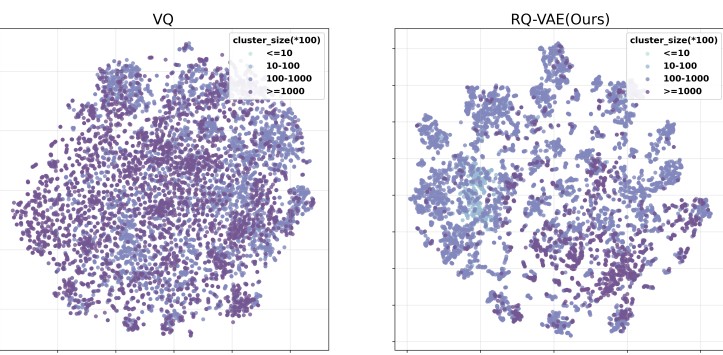

Figure 9: Visualization of the distribution of the 1st level codebook.

**SID Visualization.** Figure 10 offers an intuitive insight into the hierarchical structure of semantic identifiers generated after optimization under RQ-VAE. **i)** The semantic information on which the generation process relies ensures that items sharing the same SID have similar or identical meanings. **ii)** The residual-based computation results in a weaker semantic distinction at the last level of the hierarchical codebook, where products mapped to different third-level codebooks (e.g., $C_14C_24328C_3859$ and $C_14C_24328C_31623$) show semantic consistency. **iii)** Codebooks at earlier levels have a greater impact on semantics. Changing the 2-level codebook shifts the item category to clothing, while remaining within the drawer class. However, modifying the first-level codebook directly switches the item to an electronic device. These changes are consistent with our multimodal feature-based residual quantization generation.

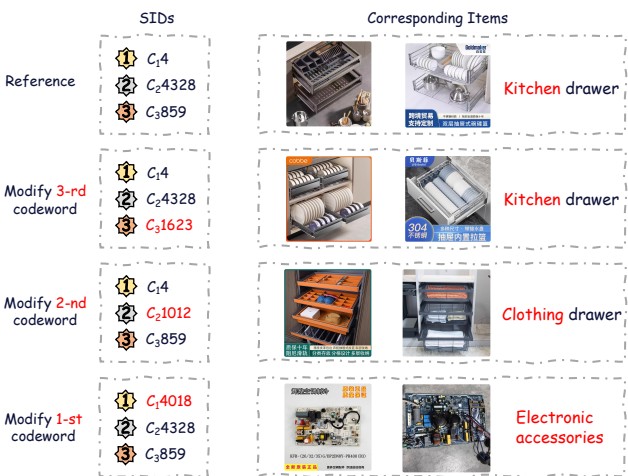

Figure 10: Visualization of corresponding items under modification to the SID.

### A.10 NOTATIONS

All the notations used in this paper can be found in Table 21.

Table 21: Summary of Notations.

| Symbol | Description |
|---|---|
| **Item Features** | |
| $\mathcal{I}_{\text{text}}^i$ | Text description of item $i$ |
| $\mathcal{I}_{\text{Image}}^i$ | Image of item $i$ |
| $i^+$ | The item that owns the most collaborative relations with item $i$ |
| $i^-$ | Items collected with in-batch negative sampling |
| **Feature Extraction** | |
| $\mathcal{M}_{\text{text}}$ | Text Encoder |
| $\mathcal{M}_{\text{image}}$ | Image Encoder |
| $\mathcal{M}_{\text{Fusion}}$ | Model used to Aggregate Multi-modal Features |
| $\mathcal{H}^i$ | Extracted multi-modal features of item $i$ |
| **Item Tokenization** | |
| $\mathcal{E}$ | Encoder of SID generation |
| m | Total level of the SIDs |
| $z_1, ..., z_m$ | Input for codebooks of each level |
| $c_1, ..., c_m$ | Identifier/Codebook of each level |
| $\mathcal{D}$ | Decoder for reconstruction |
| **Loss Function** | |
| $\mathcal{L}_{\text{cont}}$ | Contrastive loss for alignment between relative items |
| $\mathcal{L}_{\text{recon}}$ | Reconstruction loss to retain semantic information |
| $\mathcal{L}_{\text{rec}}$ | The next-token predition loss for retrieval |

