# OpenReview forum: "FORGE: Forming Semantic Identifiers for Generative Retrieval in Industrial Datasets"
_ICLR.cc/2026/Conference — Submitted to ICLR 2026_

### Official Review · Reviewer_4riH · 2025-10-19

**Soundness:** 4
**Presentation:** 3
**Contribution:** 3
**Rating:** 6
**Confidence:** 4

**Summary:**

This paper proposes FORGE, a benchmark for semantic identifier (a.k.a., semantic ID) in generative retrieval with industrial datasets. In FORGE, the recorded user interactions and items exceed billion scale and million scale respectively. Moreover, FORGE proposes the embedding hitrate and introduces Gini coefficient to evaluate the quality of semantic IDs, without time-consuming optimizations of generative recommender systems. Last but not least, FORGE provides several insights towards how to transfer a collection of well-trained semantic IDs into real-world production systems. To sum up, FORGE is a solid work that lays a foundation for the further development of generative recommendation systems.

**Strengths:**

1. An industrial dataset for information retrieval tasks, including recommendation, search, and etc., is desirable while demanding. FORGE provides a solid benchmark for the development of the next-generation recommender systems.
2. The embedding hitrate proposed in this work is intriguing and practical, which can mitigate the evaluation problem of semantic IDs. Previously, we do need to first finish the complete training process of a recommender system with large-scale parameters, and then we are able to figure out the IDs quality indirectly, according to the recommendation performance.
3. The strategy that transfers the well-behaved IDs to online systems is valuable.
4. The experimental results of this work is expensive and meaningful.

**Weaknesses:**

1. According to this manuscript, FORGE merely releases the continuous embeddings, instead of the raw attributes including texts and images. Despite of the privacy considerations, frozen embeddings may restrict the flexibility of what the community can carry forward based on FORGE. As far as I know, current leading generative recommender systems tend to customize their own semantic IDs from the raw features.
2. Some contributions are overstated. If I understand correctly, the optimization strategies of semantic ID generative actually refer to the KNN-based and Random-based strategies in Section 3.2, which are designed to mitigate ID collision.
3. The most significant improvement of performance benefits from the Random-based strategy, compared with the KNN-based strategy and the incorporation with side information, while lacks in-depth discussion and analysis.
4. As to the embedding hitrate,  are the recommendation results retrieved based on the similarity between feature embeddings, or semantic IDs embeddings?

**Questions:**

1. Compared with traditional recommender systems based on user-item inner-product, are generative recommender systems more computationally efficient?
2. As for as I know, generative recommender systems start to develop novel semantic IDs, including dynamic IDs [1], parallel IDs[2], and achieve some progresses. I wonder if the authors have further plans to investigate these fancy techniques.
3. I wonder the collision rate of different quantization techniques when generate semantic IDs for item collection of million scale.

[1]. Unleash LLMs Potential for Sequential Recommendation by Coordinating Dual Dynamic Index Mechanism. WWW 2025.

[2]. Generating Long Semantic IDs in Parallel for Recommendation. KDD 2025.

---

> ### Author Response · Authors · 2025-11-22
> **Rebuttal by Authors**
>
> Thank you for your comprehensive review and insightful feedback.
>
> `W1: According to this manuscript, FORGE merely releases the continuous embeddings, instead of the raw attributes, including texts and images. Despite the privacy considerations, frozen embeddings may restrict the flexibility of what the community can carry forward based on FORGE. As far as I know, current leading generative recommender systems tend to customize their own semantic IDs from the raw features.`
> - **Explore end-to-end training from raw attributes is currently not feasible in industrial settings due to efficiency constraints.**
>     - On our platform, billions of user interactions are generated daily, and the model must be updated daily based on these interactions. **Even with 160 GPUs, it already takes approximately 15 hours per day to train the recommenders**. Incorporating large-scale multimodal models would significantly hinder the model’s ability to iterate and adapt quickly.
>     - In line with this practical consideration, we emphasize that end-to-end modeling [1] does not necessarily have to start from raw features. **We are currently exploring the end-to-end schema with pre-extracted multimodal embeddings, as it only brings little disruption to the regular iteration of recommenders.** Since this paper mainly focuses on static SID and downstream GRs, we leave this aspect for future work.
> - **The wide range of usage of FORGE.**
>     - **Semantic Identifiers Generation:** With the provided multi-modal features in `AL-GR/Item-Info`, users could construct their SID with different configurations (e.g., different SID levels, SID postprocessing, SID Encoding, etc.) and subsequently assess the quality of SIDs using our proposed two new metrics.
>     - **Generative recommendation/Search with semantic identifiers:** Researchers could further evaluate the effectiveness of their constructed SID with both recommendation and search data we provided in `AL-GR/AL-GR-v1`. Moreover, they could also perform a thorough comparison between their proposed models with other SOTA baselines using our datasets.
>     - **Standard retrieval tasks using single token identifiers.** As we also release the original user interaction sequences in `AL-GR/Origin-Sequence-Data` using single token identifiers, it would also be convenient for users to perform traditional retrieval as a real-world benchmark.
>
> `W2: Some contributions are overstated. If I understand correctly, the optimization strategies of semantic ID generative actually refer to the KNN-based and Random-based strategies in Section 3.2, which are designed to mitigate ID collision.`
>
> We would like to clarify that our “SID optimization strategies” in this paper comprise a set of design dimensions that we systematically **investigate and validate**, including:
> - multimodal and collaborative data used in SID generation (Section 3.1, Tables 6 and 16),
> - ID collision mitigation strategies (we proposed in Section 3.2, Tables 2 and 15),
> - quantization methods and SID level structures (Appendix A.7, Tables 17 and 18),
>
> Through extensive and detailed experiments, we found that input data modality, different quantization methods, and ID collision mitigation strategies are closely related to the performance of generative recommendation tasks, while the SID hierarchical structure has a minimal impact on recommendation performance. This guides subsequent research in the field of generative semantic ID. **To avoid ambiguity, we have replaced the "SID optimization strategies" with "investigates several strategies for SID generation" in the revised version**.

---

> ### Author Response · Authors · 2025-11-22
> **Rebuttal by Authors (continued)**
>
> `W3: The most significant improvement of performance benefits from the Random-based strategy, compared with the KNN-based strategy and the incorporation with side information, while it lacks in-depth discussion and analysis.`
> - First, we have to claim that KNN and Random primarily fall under the **post-processing strategies for SID conflict avoidance,** while i2i and sideinfo could be classified as **data modality.** These two aspects are parallel and do not interfere with each other, so the random-based methods can be combined with the i2i- and sideinfo-based methods (Random-5 + i2i + sideinfo = Ours).
> - Second, KNN and Random both aim to improve accuracy and reduce the ID collision rate by dispersing the third-layer SIDs. The difference lies in that KNN assigns sub-optimal codewords while still preserving some semantic information, whereas Random focuses on achieving the highest possible accuracy, which leads to a lower Gini coefficient in Figure 3(b). From the results, Random performs slightly better than KNN, and we believe this is because the semantic information retained in the final codeword is less critical compared to precision and diversity due to the constraint of residuals in RQ-VAE.
>
> `W4: As to the embedding hitrate, are the recommendation results retrieved based on the similarity between feature embeddings or semantic ID embeddings?`
>
> It is calculated with the recommendation results retrieved based on the similarity between feature embeddings. The calculation process could be found in Section 3.4.
>
> `Q1. Compared with traditional recommender systems based on user-item inner-product, are generative recommender systems more computationally efficient?`
>
> We analyze computational efficiency from two perspectives:
> - **Hardware Utilization:** The Qwen2.5-0.5B-based recommender achieves 124.32 TFLOPS, far exceeding SASRec's 0.85 TFLOPS. This is because its dense Transformer layers (0.35B parameterized operations) fully engage GPU tensor cores, making it compute-bound.
> - **Per Inference Computational Overhead:** Under the same QPS and hardware, its latency is ~10× higher than traditional vectorized retrieval models. This stems from autoregressive generation and the need to evaluate a large network with billions of parameters at each step, preventing caching or parallel generation.
>
> Thus, while generative recommenders achieve high hardware utilization, they are less computationally efficient per inference due to high sequential computation cost.
>
> `Q2. As far as I know, generative recommender systems start to develop novel semantic IDs, including dynamic IDs [1], parallel IDs[2], and achieve some progress. I wonder if the authors have further plans to investigate these fancy techniques.`
>
> We have implemented OPQ (parallel IDs [2]) in the following Table. More analysis about the dynamic IDs [1] can be found in W1 as we have mentioned.
> |Method|HR20|HR100|HR500|HR1000|
> |---|---|---|---|---|
> |OPQ|3.57%|9.57%|21.19%|26.64%|
> |Ours|4.89%|12.31%|24.79%|29.01%|
>
>
> `Q3. I wonder the collision rate of different quantization techniques when generate semantic IDs for item collection of million scale (900M).`
>
> In our experiments, with approximately 900 million items and all other configurations held constant, we evaluated the collision behavior of various quantization methods using a collision threshold of 25 (i.e., when the number of items mapped to a single SID exceeds a predefined threshold, all exceeding items associated with that SID are treated as conflicting items). The observed collision rates are as follows: RQ-VAE: 10%, VQ (Vector Quantization): 80%, RQ-kmeans: 35%.
> |Method|collision rate|number of collision item|
> |---|---|---|
> |Multi-VQ|78.79%|709,107,518|
> |RQ-kmeans|35.25%|317,270,826|
> |RQ-VAE|10.17%|91,522,043|
>
>
> [1]. Unleash LLMs Potential for Sequential Recommendation by Coordinating Dual Dynamic Index Mechanism. WWW 2025.
>
> [2]. Generating Long Semantic IDs in Parallel for Recommendation. KDD 2025.

---

> > ### Comment · Reviewer_4riH · 2025-11-23
> >
> > First, thanks for your response.
> >
> > > Weakness 1.
> >
> > I fully understand the difficulty of end-to-end training from raw attributes in industrial settings. However, the raw attributes, i.e., textual contents and visual images, are more beneficial to the research community, compared with the frozen embeddings. Researchers and practitioners are allowed to customize their own recommender systems from the attributes level, which is regardless of the end-to-end training.
> >
> > > Question 3.
> >
> > Please clarify the detail of item SID generation in *the evaluated OPQ*. Does it still adopt auto-regressive generation?
> >
> >
> > > Question 4.
> >
> > I wonder if a collision rate that is larger than 30% (Multi-VQ and RQ-kmeans) is practicable in real-world systems?

---

> ### Author Response · Authors · 2025-11-23
>
> Thank you for your insightful comment.
> > Weakness 1.
>
> We appreciate your point that “raw attributes (i.e., text content and images) are more beneficial to the research community.” However, releasing the raw data for all 250 million items and 14 billion user interactions would pose significant privacy risks to both users and our company. For instance, third parties could potentially reconstruct user behavior trajectories and competing e-commerce platforms might infer detailed user profiles specific to our platform. Such disclosures would violate the privacy commitments we make to users under our platform’s terms of service.
>
> > Question 3.
>
> Our work primarily investigates the impact of SID encoding on GR tasks. To ensure a fair comparison and maintain compatibility with our internal engineering infrastructure, we adopt only OPQ’s encoding method while relying on autoregressive generation combined with dynamic beam search for GRs.
>
> We acknowledge that applying autoregressive generation on OPQ for GR may underestimate its performance. To avoid misleading readers, we have explicitly clarified this in the revised version (Lines 1131–1133).
>
> > Question 4.
>
> For the experiment in Question 4, we do not employ any alleviating collision techniques to validate the ID collisions.
>
> About the collision rate:  When the number of items mapped to a single SID exceeds a predefined threshold (25 in the rebuttal), all exceeding items associated with that SID are treated as conflicting items. The collision rate is defined as:
>
> `Collision Rate = Number of Conflicting items / Number of Total items`
>
> We are evaluating the performance of RQ-Kmeans and Multi-VQ on GRs **without any alleviating collision strategy** to validate whether their settings are practicable in real-world systems. The training of GRs will take approximately two days and we will provide the results via comments once it is completed. (The results of the RQ-Kmeans and VQ methods reported in Table 18 are already based on the KNN-based strategy for SID collisions)

---

> ### Author Response · Authors · 2025-11-25
>
> Thanks again for taking the time to provide additional feedback and share your concerns with us.
>
> > Question 4. Whether a collision rate that is larger than 30% (Multi-VQ and RQ-kmeans) is practicable in real-world systems.
>
> We have finished the training of GRs using the SIDs from OPQ, Multi-VQ and RQ-KMeans **without any collision mitigation strategy**. The results are summarized in the following table, where a larger collision rate actually leads to a smaller HitRate (e.g., HitRate: Multi-VQ < OPQ < RQ-Kmeans < RQ-VAE), demonstrating that the collision rate plays a critical role in industrial deployment.
>
> |method| HR20 | HR100 | HR500 | HR1000 |collision_rate|
> |---|---|---|---|---|---|
> |Multiple-VQ (w/o collision mitigation)|0.42%|1.34%|4.11%|6.59%|78.79%|
> |OPQ (w/o collision mitigation)|1.31%|3.52%|8.95%|12.76%|55.20%|
> |RQ-Kmeans (w/o collision mitigation)|1.99%|5.50%|13.67%|18.97%|35.25%|
> |RQ-VAE (w/o collision mitigation)|2.96%|7.69%|17.63%|22.27%|10.17%|
>
> We also welcome any further questions or discussions and appreciate the opportunity to enhance our work.

---

> > ### Comment · Reviewer_4riH · 2025-11-26
> >
> > Thanks for your response.
> >
> > > Question 4. Whether a collision rate that is larger than 30% (Multi-VQ and RQ-kmeans) is practicable in real-world systems.
> >
> > Could you further elaborate on the evaluation details? The presented result is a little bit counterintuitive. Since collision indicates that some distinct items share the same semantic id, this issue may result in a spuriously high hitrate. Extremely, if all items are collided, the hitrate can trivially achieve 100%.

---

> ### Author Response · Authors · 2025-11-26
>
> Thank you for raising this insightful concern.
>
> > Question4. Could you further elaborate on the evaluation details?
>
> We would like to clarify that all evaluation metrics reported in our paper are computed **at the item level**, rather than the SID level. Therefore, metrics like HitRate@1000 would **retain the Top 1000 items as the retrieval results**.
>
> Specifically, the pipeline is described in the repository `algr/calc_hr.py`. Taking a 3-level SID with 8192 codebooks per level as an example:
> - We first generate a list of 1200 candidate SIDs based on the beam search of GR
> - Then, we **map these SIDs back to their corresponding original items** (e.g., the candidate list: (SID1) $item_1^1$, $item_2^1$, ..., $item_{I_c(SID1)}^1$, ..., (SID1200) $item_1^{1200}$, $item_2^{1200}$, ..., $item_{I_c(SID1200)}^{1200}$, where SID1200 and $I_c(SID1200)$ denote the SID with a rank of 1200 and the number of items corresponding to SID1200)
> - Finally, we compute HitRate over the resulting item list by **selecting the top-k items with the highest scores** and checking whether each of them matches the ground truth items in the session.
>
> `Note`: When de-quantizing SIDs, if a SID ($C_1 1248 C_2 6475 C_3 4821$) maps to M items but most slots have been occupied by those items whose SIDs have higher scores than $C_1 1248 C_2 6475 C_3 4821$, and only $n\lt m$ slots remain in Top-K, we randomly select n items within $C_1 1248 C_2 6475 C_3 4821$ into the retrieval results.  **This evaluation pipeline exactly mirrors our online inference workflow** (only the Top-K items will enter the following ranking stage), ensuring a fair and realistic assessment of model performance.
>
> The procedure ensures that even if multiple distinct items share the same SID, they are still evaluated individually in the final candidate list. As a result, performance won't be enhanced by collisions.
>
> We have added a detailed description of this process in Lines 282-285 of the revised version to improve clarity.
>
> > Extremely, if all items collide, the HitRate can trivially achieve 100%.
>
> In the extreme case where all items share the same SID, we still retain at most K items after de-quantization. Since generative retrieval cannot distinguish among items within the same SID, we uniformly sample up to K items from the candidate pool. As a result, the final HitRate@K on items is equivalent to that of random selection.

---

> > ### Comment · Reviewer_4riH · 2025-11-28
> >
> > First, thanks for your detailed response. From my perspective, this paper now deserves a rating higher than 6, while marginally below 8. Hence, I raise the contribution score to 4 and confidence to 5. Best wishes and good luck.

---

> > > ### Author Response · Authors · 2025-11-28
> > >
> > > We sincerely appreciate your thoughtful response and consideration of our paper. Your insightful questions have offered us a valuable opportunity to further strengthen our work.

---

### Official Review · Reviewer_kG4V · 2025-10-31

**Soundness:** 2
**Presentation:** 2
**Contribution:** 2
**Rating:** 4
**Confidence:** 3

**Summary:**

This paper introduces FORGE, a generative recommendation dataset built upon real-world industrial scenarios, and focuses on investigating the impact of different SID construction strategies on model performance.

**Strengths:**

1. The authors have made a valuable contribution by promoting the open-sourcing of a large-scale industrial dataset, which is rare and highly significant for both academia community.
2. They conducted extensive analytical experiments and shared important insights from online studies, offering deep understanding into the optimization of various components.
3. The proposed metrics， embedding hitrate and Gini coefficient， provide an inspiration for evaluating SID quality without additional training.

**Weaknesses:**

1. The overall structure of the paper requires major revision. A large portion of the manuscript focuses on SID optimization strategies (e.g., Contrastive Learning and RQ-VAE Tokenization in Sec. 3.1), which are not novel contributions of this work. In contrast, the explanations for the newly proposed metrics (embedding hitrate and Gini coefficient) are insufficiently detailed.
2. All experimental conclusions are drawn solely from the proposed FORGE dataset. The absence of validation on other existing datasets weakens the generalizability of the results. Including experiments on at least one additional dataset would substantially strengthen the soundness of the claims in this paper.
3. The paper proposes to resolve SID conflicts using KNN or random strategies, but the underlying motivation for these choices is not clearly discussed. Extending the codebook level at the begining seems a more natural solution to SID conflicts. So why is the proposed approach preferable?

**Questions:**

1.Line 269 states that FORGE employs dynamic beam search, but the paper neither provides methodological details nor cites relevant references. It is therefore unclear whether this represents a genuine technical contribution. Please provide more specifics.
2. The paper evaluates performance using HR@20, 100, 500, 1000, which implicitly suggests that the proposed method targets the recall stage of a recommendation pipeline. Were the results further processed in subsequent stages (e.g., ranking or re-ranking)?

---

> ### Author Response · Authors · 2025-11-22
> **Rebuttal by Authors**
>
> Thank you for your comprehensive review.
>
> `W1: The overall structure of the paper requires major revision. `
>
> We have revised the manuscript accordingly. Specifically, we have streamlined these sections by providing only brief introductions that clarify their roles and highlight alternative approaches used in the experiments. Besides, we have expanded the discussion in Section 3.4 and added some important experiments to improve overall clarity and focus.
>
> `W2: Including experiments on at least one additional dataset. `
>
> We have now included experiments on another widely used dataset from Amazon2023-Clothing_Shoes_and_Jewelry, which consists of 7.2M items and 66.0M interactions. We construct 3-level codewords with 512 codebooks per level (i.e., 3x512). The base configuration uses only image and text features together with a KNN-based post-processing strategy, where the collision threshold \sigma is set to 25. In contrast, Ours incorporates additional item-to-item collaborative signals and adopts a Random-based post-processing strategy with \sigma = 5. The results show consistent improvements of `ours` over `base`, demonstrating that our proposed methods remain robust on smaller-scale datasets.
> |Method|HR20|HR100|HR500|HR1000|
> |---|---|---|---|---|
> |base (3x512)|1.02%|2.47%|5.71%|8.07%|
> |ours (3x512)|2.12%|4.56%|9.79%|13.04%|
>
> `W3: The paper proposes to resolve SID conflicts using KNN or random strategies, but the underlying motivation for these choices is not clearly discussed. Extending the codebook level at the beginning seems a more natural solution to SID conflicts. So why is the proposed approach preferable?`
> - **The motivation for using KNN or random strategies:** The main idea is to preserve semantic information in the first few layers, while using the last level primarily to diversify items, as the last layer retains limited semantic information under the residual approach. Therefore, the precision of the last level is much more important than preserving semantic information.
> - **Why not simply extend the codebook with another level?**
>     - Unlike vanilla LLMs, industrial recommenders for the retrieval stage typically return thousands of items per request (large beam size) and must handle almost **ten thousand requests per second** in our platform. Adding a codebook level would require an extra beam search step (1200 more decoding steps for each request, with 1200 as the beam size ), which could significantly increase computational overhead.
>     - The performance difference between a four-layer SID and a three-layer SID without resolving SID conflicts is minimal, as shown in Table 17. In contrast, keeping the 3-level SID and applying our KNN or Random strategy yields substantial gains (e.g., up to +15–20% relative HR@20 over the base configuration across S1–S3 in Table 2), without increasing the number of decoding steps.
>
> `Q1: Please provide more specifics of dynamic beam search.`
> - In the context of LLMs, a common approach is to retain the top-K tokens (in our case, K = 1200) after the prefilling phase. As a result, each decoding step must receive K sequences as input, leading to a computational cost that is approximately 1200 times. This makes it challenging to handle a large number of requests (~10,000 qps in our platform) efficiently in online scenarios.
> - To address this issue, we implemented an engineering optimization by controlling the beam size dynamically across decoding steps. Specifically, we set the beam size to 300 for the first decoding step and 600 for the second, while keeping it at 1200 only for the final decoding step. This way, the first decoding step retains only 300 sequences, and the second decoding step processes significantly fewer sequences compared to the non-dynamic approach (which would involve 1200 sequences as input). As a result, the computational cost of the second decoding step is reduced by 75% and the number of SIDs remains 1200 after the third decoding step.
>
> We have added it to the main text in the revised version (Lines 271-277).
>
> `Q2: Were the results further processed in subsequent stages (e.g., ranking or re-ranking)?`
>
> In our offline evaluation, we use Hit Rate as the primary metric to assess the performance of the retrieval model. However, in the online setting, the retrieved items will go through subsequent stages such as ranking and re-ranking before being exposed to users (Lines 482-487). User feedback is then collected based on their current interests, which includes metrics such as PVR (Page View Rate), Hit Rate, and transaction counts shown in Table 7.

---

> > ### Comment · Reviewer_kG4V · 2025-11-27
> >
> > The authors have addressed most of my concerns in their rebuttal; therefore, I will revise my scores accordingly.

---

> > > ### Author Response · Authors · 2025-11-28
> > >
> > > Thank you very much for your valuable time and effort in helping us improve our work. We are encouraged that the rebuttal has addressed your concerns.

---

### Official Review · Reviewer_ZTNg · 2025-11-01

**Soundness:** 3
**Presentation:** 1
**Contribution:** 3
**Rating:** 4
**Confidence:** 5

**Summary:**

This paper proposes FORGE, a large-scale benchmark dataset designed for generative recommendation in industrial environments. The authors identify three major challenges in existing generative recommendation research:
1.Limited data scale — existing datasets are too small for training generative recommenders at realistic scales.
2. Neglect of semantic ID generation — the lack of semantically meaningful item identifiers constrains generative modeling.
3. Inefficient industrial deployment — current generative recommenders struggle with latency and scalability in practice.

To address these challenges, the authors release the FORGE benchmark, a large multimodal dataset with text, image, and behavioral features. The dataset is impressive in scale, containing: 100× more user interactions than the largest existing public recommendation dataset, 26× more users, and 3× more items.

Additionally, the paper discusses a semantic ID generation framework and industrial inference acceleration strategies to make generative recommendation feasible at scale.

**Strengths:**

1. The release of a large-scale, multimodal, industry-grade dataset fills a critical gap in the generative recommendation literature. The dataset has clear potential to become a standard benchmark in this area, especially when comparing with current small recommendation datasets.

2. By including textual and visual item representations, the dataset supports training and evaluation of multimodal generative recommenders that are closer to real-world systems.

3. The work is timely given the rise of semantic ID representations, originating from Differentiable Search Index (DSI) models and later adapted for generative retrieval and recommendation. The authors’ emphasis on semantic tokenization connects well with current progress in this field.

**Weaknesses:**

1. Writing and Organization
The paper’s readability needs substantial improvement. The objectives are not clearly separated: it is unclear whether the focus lies on the dataset release or the proposed generative recommendation framework. Please consider clarify the contributions splitting the paper into two parts:
(1) Dataset and benchmark release, including comparisons across multiple settings and baselines;
(2) The proposed generative recommendation framework with technical details and industrial deployment results.
The introduction currently mixes dataset and model contributions, making it difficult to link back to the three challenges stated early in the paper. Each challenge should be explicitly addressed in corresponding sections.

2. Missing Details on Semantic ID and Tokenization

The paper should provide more clarity and technical depth on Semantic ID formation: The authors employ RQ-VAE, which has become standard since TIGER, but the paper lacks discussion on: 1. How RQ-VAE was adapted for multimodal features in FORGE, 2. How gradient updates are controlled during training, 3. Whether the authors use the Straight-Through Estimator (STE) strategy, as in “Generative Recommender with End-to-End Learnable Item Tokenization” (SIGIR 2025), how did the author address the so called "dead code" problem in the codebook tokenization procedure? Whether the learned codes differ in interpretability or compositionality from prior works.
Including an ablation study or visualization of the semantic ID space would strengthen the paper significantly. etc.

3. Related Work is Incomplete,  The related work section requires major expansion and reorganization. The paper should contextualize its contributions in relation to generative retrieval and semantic ID learning. Actually,  “Editing Factual Knowledge in Language Models” (2021) introduces the notion of editing and retrieving via generation. Then the generative retrieval has been succssfully applied to both search and recommendation domain. The authors focus on the recommendation domain, neglect several relevant studies in search domain. For example,

3.2. Tokenization and Generative Search Advances
a. Learning to Tokenize for Generative Retrieval, NeurIPS 2023.
b. Scalable and Effective Generative Information Retrieval, WWW 2024.
3.3. Generalization in Generative Retrieval
a. Replication and Exploration of Generative Retrieval over Dynamic Corpora, SIGIR 2025.
b. Constrained Auto-Regressive Decoding Constrains Generative Retrieval, SIGIR 2025.
3.4. Bridging Search and Recommendation
a. Bridging Search and Recommendation in Generative Retrieval: Does One Task Help the Other? (2024).
b. Semantic IDs for Joint Generative Search and Recommendation (2025) — which also employs the Gini index and should be cited.
c. Unifying Search and Recommendation: A Generative Paradigm Inspired by Information Theory (2024).

Please consider introducing a dedicated subsection in Related Work to systematically discuss these categories (foundations, tokenization, generalization, and task unification). This would clarify the position of FORGE within the broader generative retrieval/recommendation research landscape.

4. Clarity of Benchmark Evaluation
The benchmark results lack clarity in explaining performance differences across semantic ID configurations. Since RQ-VAE and similar tokenizers have become standard practice, the paper should explicitly show:
4.1.  Ablation studies across multiple semantic ID variants,
4.2. Cross-modal performance breakdowns (text, image, and user interaction modalities),
4.3. The contribution of tokenization quality to overall recommendation metrics.

Without this, it remains unclear how the proposed semantic ID learning improves over existing methods.

5. Missing Online Evaluation

The paper currently reports offline results only. Since the authors claim industrial applicability, online evaluation (e.g., A/B testing or simulation) is crucial to assess real-world effectiveness.

Please consider adding a figure or table summarizing online evaluation or latency-performance trade-offs from industrial deployment.

6. Multimodal Feature Impact

Although FORGE includes multimodal data, the paper does not analyze how multimodal features (e.g., visual vs textual) affect model performance.

Probably the authors can add a subsection on multimodal ablation — e.g., comparing text-only, image-only, and joint input setups — to demonstrate the impact of multimodality.

7. Reference Redundancy

Several references are redundant or inconsistent (e.g., the same paper appears as 2024b and 2024c). These should be carefully cleaned and deduplicated before submission.

**Questions:**

1. About the RQ-VAE usage in the proposed method：The authors employ RQ-VAE, which has become standard in generative recommednation since TIGER, but the paper lacks discussion on: 1. How RQ-VAE was adapted for multimodal features in FORGE, 2. How gradient updates are controlled during training, 3. Whether the authors use the Straight-Through Estimator (STE) strategy, as in “Generative Recommender with End-to-End Learnable Item Tokenization” (SIGIR 2025), how did the author address the so called "dead code" problem in the codebook tokenization procedure? Whether the learned codes differ in interpretability or compositionality from prior works.
Including an ablation study or visualization of the semantic ID space would strengthen the paper significantly. etc.

2. Online evalution: The paper currently reports offline results only. Since the authors claim industrial applicability, online evaluation (e.g., A/B testing or simulation) is crucial to assess real-world effectiveness.

I expect the authors to address the questions and resolve the issues listed above and the weaknesses. I will therefore revise my score if they are able to satisfactorily address my concerns.

---

> ### Author Response · Authors · 2025-11-22
> **Rebuttal by Authors**
>
> We sincerely appreciate the constructive feedback provided.
>
> `W1: Writing and Organization.`
>
> Thanks for your suggestion regarding the writing and organization of the paper. In response, we have revised the manuscript to clarify the structure and emphasize the main contributions.
>
> Our work primarily focuses on the **dataset and benchmark** for generative retrieval with semantic identifiers (SIDs), rather than introducing an entirely new generative recommendation architecture. The generative recommendation framework is already widely adopted. Within this established framework, our technical focus lies in understanding **how SIDs are formed and evaluated** and **how design choices impact performance in real-world industrial systems.**
> - **Dataset.** As the reviewer mentioned, we released a large-scale industrial dataset specifically tailored for generative retrieval with SIDs, which contains 14 billion user interactions and multimodal features of 250 million items. To the best of our knowledge, this is the first public benchmark of such scale for SID-based GR.
> - **SID generation strategies and direct evaluation.** Using this dataset, we systematically investigate the impact of several SID generation strategies on recommendation performance, including:
>     - data modalities (text, image, i2i)
>     - SID quantization methods and level structures
>     - SID collision-handling strategies (e.g., KNN-based and Random-based)
>     - We evaluate these strategies extensively through both offline experiments and online A/B tests. In addition, we propose two novel metrics for direct SID evaluation (i.e., embedding hitrate and the Gini coefficient) that correlate strongly with downstream GRs and allow SID assessment without any GR training.
> - **Efficient deployment and convergence.** We further study a practical pre-training strategy based on two days of offline user-action pretraining. This warm-up reduces the online convergence time of the new SIDs from about 15 days to 7 days in our production system, making it easier to validate SID improvements in real-world deployments.
>
> To improve the paper's readability, we have reorganized the Introduction section and explicitly introduced labels for Challenge 1, Challenge 2, and Challenge 3 to directly align with the above contributions.
>
> `W2&Q1: Missing Details on Semantic ID and Tokenization. How did the author address the so-called "dead code" problem in the codebook tokenization procedure? Including an ablation study or visualization of the semantic ID space would strengthen the paper significantly.`
> - **Details on Semantic ID and Tokenization.**
> We clarify the technical details of Semantic ID (SID) formation in FORGE as follows:
> As shown in Figure 1, SID generation consists of two sequential stages: (1) multimodal feature fusion and (2) standard RQ-VAE tokenization.
>     - **Adaptation to Multimodal Features:** In the first stage, we use a CLIP-style pretrained encoder to map heterogeneous multimodal features (e.g., text, image) into a unified semantic embedding space. This fused embedding serves as the input to the RQ-VAE in the second stage.
>     - **Gradient Update Control:** The two stages are trained separately. We first optimize the multimodal fusion module to produce stable and expressive embeddings, then fix this encoder and train the RQ-VAE independently. This staged training decouples the optimization dynamics, avoiding unstable gradient propagation from the discrete codebook learning back into the early fusion layers
>     - **Use of Straight-Through Estimator (STE):** Since the RQ-VAE is trained after the fusion stage with the encoder frozen, there is no need to propagate gradients through discrete codes to the multimodal encoder. Therefore, while we do use STE within the RQ-VAE’s codebook update (as standard), we do not require the extended STE strategy proposed in prior work for end-to-end joint training.
> - **Dead Code: The "dead code" issue arises from (1) codebook collapse and (2) ID collisions due to similar samples clustering.**
>     - **For codebook collapse,** we initially used RQ-KMeans initialization. But we also found that with high-quality and discriminative embeddings, even random initialization can achieve uniform code usage, making collapse negligible in practice.
>     - **For ID collisions,** we propose KNN-based and random-based post-strategies to reduce ambiguous mappings, as detailed in Section 3.2.
> - **Ablation study or visualization.**
> We have added t-SNE visualization and ablation on different quantization techniques in the revised version (Appendix A.7), where RQ-VAE exhibits stronger discriminability and lower collision rate compared to VQ-based encoding.

---

> ### Author Response · Authors · 2025-11-22
> **Rebuttal by Authors (continued)**
>
> `W3: Related Work is Incomplete. The related work section requires major expansion and reorganization.`
>
> We have now added them in the revised version.
>
> `W4: The paper should explicitly show: 4.1. Ablation studies across multiple semantic ID variants, 4.2. Cross-modal performance breakdowns (text, image, and user interaction modalities), 4.3. The contribution of tokenization quality to overall recommendation metrics. `
> - **Ablation studies across multiple semantic ID variants.** The ablation studies about different ID collision, different data modalities, different SID level structures, and different quantization methods are **placed in tables 14, 15, 16, and 17 during the initial submission**. In these tables, we have thoroughly investigated the impact of different semantic ID variants on the retrieval performance.
> - **Cross-modal performance breakdowns.** We conducted more ablation experiments on the data modalities used in the multimodal representation. As shown in the table (Lines 466 - 479 in the revised version), using only text and image for representation generation leads to a significant performance drop. In comparison, the i2i-based method is more effective for recommendation tasks, but it still underperforms compared to the fusion of multiple modalities.
> |Stage|Method|HR20|HR100|HR500|HR1000|
> |---|---|---|---|---|---|
> ||text-only|3.01%|7.97%|18.00%|21.01%|
> |S1|image-only|3.06%|8.34%|18.79%|22.03%|
> ||i2i-only|4.63%|11.58%|22.52%|23.15%|
> ||our|4.89%|12.31%|24.79%|29.01%|
> - **The contribution of tokenization quality to overall recommendation metrics** can be observed from Figure 3, which states that embedding hitrate and gini coefficient (for SID conflicts) play a more crucial role in recommendation. Figure 7 also emphasizes that some metrics, like Feature Fidelity, have less relation to recommendations.
>
> `W5&Q2: Missing Online Evaluation.`
>
> **The online assessment of our framework has been placed in the initial submission and can be found in Table 7,** which shows the improvement through online experiments deployed on the homepage of our platform for recommendation. Ours with the best configuration (KNN-5 + i2i + sideinfo) through offline validation improves the PVR and HR by 8.93% and 10.02% compared to base. Moreover, replacing base with ours ultimately results in a 0.35% absolute increase in user transactions. More details about the deployment have been included in the revised version (Lines 485-488).
>
> `W6: Multimodal Feature Impact.`
>
> Related experiments can be found in W4.
>
> `W7: Reference Redundancy.`
>
> Thanks for pointing this out. Redundant reference has been resolved in the revised version.

---

> > ### Comment · Reviewer_ZTNg · 2025-11-23
> >
> > The authors have addressed most of my concerns in their rebuttal, I will thereby revise my scores.

---

> > > ### Author Response · Authors · 2025-11-23
> > >
> > > Thank you once again for your valuable time and thoughtful review. We sincerely appreciate the insightful suggestions that have helped us improve the paper.

---

### Official Review · Reviewer_JeAR · 2025-11-10

**Soundness:** 3
**Presentation:** 2
**Contribution:** 3
**Rating:** 4
**Confidence:** 4

**Summary:**

This paper proposes a large-scale dataset for forming semantic identifiers (SIDs) for generative retrieval and recommendation. It studies a pipeline that fuses text and image embeddings while controlling collisions, and reports both offline and limited online gains. It also introduces new proxy SID-quality metrics designed to assess downstream performance without expensive model training.

**Strengths:**

1.	The proposed data is large, diverse, timely, and includes useful image information. This can be a valuable resource for the community, for both generative recommendation and other related research problem.

2.	The studied problem is important and practical.

3.	Empirical results suggest utility in realistic settings.

**Weaknesses:**

1. The approach mainly combines existing components (RQ tokenization, text + image embeddings, last-token de-duplication); the technical contribution is limited.

2. The method description lacks clarity: it is unclear what “base” includes and how multiple modalities are fused.

3. There is no comparison to strong baselines, such as classical retrieval or recommendation methods, or other generative recommenders.

4. The paper should better justify how the dataset is intended to be used, given its large scale and the absence of publicly available meta-information. For example, if the raw text and images are not released, it is unclear how others could reproduce the model training in Eq. 2.

**Questions:**

1. Can you describe what specifically constitutes the base model, how the multimodal fusion is performed, and how the side information is combined with i2i?

2. How does the generative retrieval model compare to vector retrieval and vector retrieval with vector quantization?

---

> ### Author Response · Authors · 2025-11-22
> **Rebuttal by Authors**
>
> We appreciate your thorough review and valuable insights regarding our paper.
>
> `Summary 1: Limited online gains`
>
> - **Deployment setting.**
> FORGE was deployed in the homepage recommendations of our platform, which serves around 300 million users per day. In industrial applications, there are multiple retrieval paths. Thus, we only replaced one of them with FORGE-based GRs and calculated the incremental gains it brought.
>
> - **Magnitude of improvement.**
> Under the same model architecture and training paradigm (we only change SID-related strategies), we observe **a 0.35% increase in transaction count and 10.02% increase in HitRate**, which is considered significant in large‑scale commercial systems. For reference, RankMixer [1] at ByteDance reports about a 0.3% gain in active days, and HSTU [2] of Meta yields around a 0.5% improvement when replacing the ranking stage. Our 0.35% lift is of a similar order of magnitude, but achieved by optimizing semantic IDs rather than redesigning the entire retrieval–ranking pipeline.
>
> `W1: The approach mainly combines existing components (RQ tokenization, text + image embeddings, last-token de-duplication); the technical contribution is limited.`
>
> Although our approach utilizes existing components, this is not the main focus of our work. Instead, our paper mainly focuses on the research of **datasets and benchmarks in generative retrieval with semantic identifiers and provides practical guidance for building and deploying GR**:
> - **FORGE releases a large-scale dataset for both industry and research.** FORGE contains about 100 times more user interactions than the largest existing public recommendation dataset, as well as 26 times more users and 3 times more items.
> - **We systematically benchmark which types of SIDs are better suited for generative retrieval** on large‑scale data, from multiple dimensions:
>     - data modalities (item‑to‑item, query-to-item, text, image),
>     - SID quantization methods (RQ‑Kmeans, RQ‑VAE, OPQ, etc.) and SID structures,
>     - SID collision handling (for which we introduce KNN‑based and Random‑based strategies).
>     - **We also propose two new metrics** (i.e., embedding hitrate and Gini coefficient) for directly **evaluating SID quality without any GR training**, and show they correlate well with GR performance.
> - **We propose an effective offline pretraining strategy** that reduces online convergence time from about 15 days to 7 days in our production system, making SID upgrades more practical in real‑world deployments.
>
> `W2&Q1: The method description lacks clarity: it is unclear what “base” includes, how multiple modalities are fused, and how the side information is combined with i2i?`
> - **The definition of “Base”.**
> It can be found on lines 323-325 in the original paper (lines 325-327): "Base is constructed using limited item-to-item collaborative information as defined in Equation 2, along with a KNN-based post-processing where the threshold $\sigma$ is set to 25."
> - **How multiple modalities are fused.**
> We use a multimodal encoder based on CN‑CLIP: text features are extracted by $M_{text}$, image features by $M_{image}$, and then fused by a QFormer‑style fusion module $M_{Fusion}$ (Eq. (1)). We have clarified this fusion process in Section 3.1 with more explicit wording and also provided a minimal working demo in our anonymous repository to clearly illustrate the multimodal fusion pipeline (see `mm_clip/CLIP_demo.ipynb` in the repository), making it easier for readers to reproduce the fusion process.
> -  **How the side information is combined with i2i.**
>     - For +i2i, the positives $i^+$ are taken from the provided item‑to‑item (i2i) co‑occurrence graph, and negatives are sampled in‑batch; The contrastive training injects the collaborative signals into the final multimodal embedding, so that the learned item representations capture both content and co-occurrence information.
>     - For +sideinfo, we encode additional attributes (e.g., category, seller) into embeddings, concatenate them with $H_{text}$, and feed the concatenated vector into the same contrastive learning objective, so that both the multimodal content and side information jointly shape the item representation before quantization.
> We have now emphasized them in the revised version with a more detailed explanation and bold text.

---

> ### Author Response · Authors · 2025-11-22
> **Rebuttal by Authors (continued)**
>
> `W3&Q2: There is no comparison to strong baselines, such as classical retrieval or recommendation methods, or other generative recommenders.`
> - **For the generative recommendation.** As our primary focus is benchmarking aspects related to SIDs, we have previously compared Multiple-VQ and RQ-Kmeans implemented in Onerec [3] in the original version. In this revised version, we also incorporated OPQ [4] as shown in the following table:
> |Stage|Method|HR20|HR100|HR500|HR1000|
> |-|-|-|-|-|-|
> ||Multi-VQ|2.87%|8.15%|18.30%|23.84%|
> ||RQ-kmeans|3.42%|9.39%|20.71%|26.11%|
> |S1|OPQ|3.57%|9.57%|21.19%|26.64%|
> ||base|3.61%|9.67%|20.82%|25.09%|
> ||Ours|4.89%|12.31%|24.79% |29.01%|
>
> - **For classical retrieval.** To ensure the reliability of our results, we included several widely used baselines. The vector retrieval-based methods require computing the inner product with over 200 million item embeddings from S1, which significantly increases inference costs. To facilitate reproducibility, we sampled a smaller dataset from the test samples of S1, reducing the number of items to one-tenth of S1 (we have released it in `AL-GR/AL-GR-Tiny`). The results in the table demonstrate the effectiveness of generative retrieval with SIDs compared to **(Q2) vector retrieval-based methods and vector retrieval with vector quantization (SASRec_SID).** The code is available at `u2i/run.py`.
> |Method|HR20|HR100|HR500|HR1000|Params (Emb/Dense)|
> |-|-|-|-|-|-|
> |YoutubeDNN| 0.07%| 0.43%| 0.174%| 2.26%|1.573B/57.8k|
> |SASRec|0.15%|0.59%|1.93%|3.12%|1.573B/61.9k|
> |Bert4Rec|0.09%|0.62%|2.11%|3.26%|1.573B/61.9k|
> |SASRec_SID| 0.23%|1.24%|3.65%|5.34%|1.574B/61.9k|
> |HSTU| 0.21%| 0.77%| 2.40%|3.71%|1.573B/224k|
> |Ours|0.88%| 2.32%| 5.62%| 7.07%|0.18B/0.35B|
>
> `W4: The paper should better justify how the dataset is intended to be used. For example, if the raw text and images are not released, it is unclear how others could reproduce the model training in Eq. 2.`
>
> Specifically, our dataset offers the following applications:
> - **Semantic Identifiers Generation:** With the provided multi-modal features in `AL-GR/Item-Info`, users could construct their SIDs with different configurations (e.g., different SID levels, SID postprocessing, SID Encoding, etc.) and subsequently assess the quality of SIDs using our proposed two new metrics.
> - **Generative recommendation/Search with semantic identifiers:** Researchers could further evaluate the effectiveness of their constructed SIDs with both recommendation and search data we provided in `AL-GR/AL-GR-v1`. Moreover, they could also perform a thorough comparison between their proposed models with other SOTA baselines using our datasets.
> - **Standard retrieval tasks using single token identifiers.** As we also release the original user interaction sequences in `AL-GR/Origin-Sequence-Data` using single token identifiers, it would also be convenient for users to perform traditional retrieval as a real-world benchmark.
>
> We have incorporated the instructions in the revised version (Lines 167–180). Moreover, to facilitate the reproducibility of Eq. 2, we have provided a demo in the repository `mm_clip/CLIP_demo.ipynb`.
>
> [1] Rankmixer: Scaling up ranking models in industrial recommenders. CIKM'25
>
> [2] Actions speak louder than words: trillion-parameter sequential transducers for generative recommendations. ICML'24
>
> [3] Onerec: Unifying retrieve and rank with generative recommender and iterative preference alignment. Arxiv'25.
>
> [4] Generating Long Semantic IDs in Parallel for Recommendation. KDD'25.

---

> ### Author Response · Authors · 2025-11-28
> **We are looking forward to your further comments.**
>
> Dear Reviewer JeAR:
>
> Thank you very much for your time and thoughtful feedback. Your constructive suggestions have significantly improved the quality of our paper. We understand that you may have a busy schedule and have therefore provided a concise summary of our responses below to facilitate your quick review.
>
> - Explanation for the **improvement in online experiments** compared to other industrial methods
> - **Clarification of our contribution** from the perspective of **dataset release** and **benchmark construction** (which types of SIDs benefit GRs most, as well as some practical strategies for the SID assessment and industrial deployment)
> - **Detailed description** about the definition of "base" (Lines 328-329), the fusion of multiple modalities (Line 209) and the combination between i2i and side information (Lines 368-369)
> - **Additional experiments about the comparison with strong baselines**, including classical retrieval methods (Table 20) and other SID-based GR models (Table 18), demonstrating the effectiveness of our SID configuration
> - **More instruction** about the usage of the dataset (Lines 169-179) and another **demo in the repository** `mm_clip/CLIP_demo.ipynb` to facilitate the reproducibility of Eq. 2
>
> We sincerely hope that our responses have adequately addressed your concerns. If you have any further questions or suggestions, we remain fully open to discussion. If your concerns have been resolved, we would be grateful for your re-evaluation and favorable consideration of our revision.
>
> Thank you again for your valuable time and insightful comments.
>
> Best regards,
>
> Authors of Paper 869

---

### Author Response · Authors · 2025-12-03
**Summary of the Rebuttal Stage**

Dear PCs, SACs, ACs, and Reviewers,

We sincerely thank you for the considerable time and thoughtful effort in reviewing and discussing our submission. In response, we provide a structured summary of the key outcomes organized as follows:

---
# 1. Summary of Paper Contributions
- We release **the first industrial-scale dataset for semantic identifier-based generative retrieval with rich multimodal features.** Specifically, it contains 100 times more user interactions than the largest existing recommendation dataset, along with 26 times more users and 3 times more items.
- To enable researchers to identify which configuration benefits GRs, **we investigate and compare several strategies from multiple perspectives to enhance the SID construction.** Subsequent offline experiments and **online gains** demonstrate the effectiveness of our benchmark.
- We propose **two novel metrics** that positively correlate with GR performance, facilitating **cost-free SID quality assessment without any GR training**. Additionally, FORGE **discovers an offline pretraining strategy** that reduces the convergence time of new SIDs in production by half.
---
# 2. Highlighted Strengths from All Reviewers
- **Important Problem Setting:** The work is timely given the rise of semantic ID representations, and author's emphasis on semantic tokenization connects well with current progress in this field. (JeAR, ZTNg)
- **Dataset Releasing:** The release of a large-scale, multimodal, industry-grade dataset fills a critical gap in both generative recommendation and other related research problems. The dataset has clear potential to become a standard benchmark in this area, especially when compared with current small recommendation datasets. (JeAR, ZTNg, kG4V, 4riH)
- **A Multi-Perspective Study on SID Enhancement:** They conducted extensive analytical experiments and shared important insights from online studies, offering a deep understanding into the optimization of various components. (JeAR, kG4V, 4riH)
- **Proposing Two Useful Metrics:** The embedding hitrate and Gini coefficient proposed in this work are intriguing and practical, which can mitigate the evaluation problem of semantic IDs. (kG4V, 4riH)
- **Valuable Strategies for Online Systems:** The strategy that transfers the well-behaved IDs to online systems is valuable. (4riH, kG4V)
---
# 3. Comprehensive Addressing of All Concerns

We have addressed the following main concerns with corresponding revisions:
- **Detailed Description** (JeAR, ZTNg, kG4V, 4riH): We have introduced 1) definition of “base” and modality fusion, 2) dataset usage with both description and another demo, 3) details on Semantic ID and Tokenization, 4) related works in the search domain, 5) motivation of the SID collision strategies, 6) more specifics of the dynamic beam search, 7) online inference, 8)collision rate of different quantization techniques and 9) evaluation details about the hitrate.
- **Comparison to other baselines** (JeAR, 4riH): We added experiments on our datasets with 1) vector retrieval (also compare the efficiency), 2) vector retrieval with vector quantization, and 3) parallel SIDs.
- **Contribution Clarification** (JeAR, ZTNg, 4riH): We offer a more precise clarification of our paper’s contributions with respect to the dataset and benchmark track.
- **Paper Organization** (ZTNg, kG4V): We have revised the paper’s structure by 1) streamlining unnecessary sections while emphasizing the core contributions, 2) explicitly highlighting the objectives, and 3) removing redundant citations.
- **More studies about Benchmark Evaluation** (ZTNg): We **conducted experiments** about the cross-modal performance breakdowns and **emphasized the table in our paper** containing 1) online experiments, 2) ablation studies across multiple semantic ID variants and 3) contribution of tokenization quality to overall recommendation metrics.
- **More Datasets** (kG4V): Apart from the datasets in FORGE, we also evaluate the SID optimizations verified in our paper **on the Amazon Dataset and achieve 61.6% improvements in HitRate@1000,** thereby demonstrating the effectiveness.
---
# 4. Reviewer Ratings and Confidence

During the discussion, we are encouraged that **3 of 4 reviewers mentioned that their concerns have been addressed and increased their scores,** leading to the overall rating increase from **(6, 4, 4, 4) to (6+, 6, 6, 4)**. Specifically:
- Reviewer JeAR, with an initial rating of 4 and confidence 4, did not participate in the discussion phase.
- Reviewer ZTNg increased the presentation score to 2 and the overall rating to 6.
- Reviewer kG4V increased the overall rating to 6.
- Reviewer 4riH commented that "From my perspective, this paper now deserves a rating higher than 6, while marginally below 8, and raised the contribution score to 4 and confidence to 5".
---
We believe that our revision and replies effectively demonstrate the value of our research.

Thank you!

Best regards,

Authors of Paper 869

---

### Meta-Review · Area_Chair_H4oc · 2026-01-07

**Summary:**

This paper proposes FORGE, a large-scale industrial benchmark for forming and evaluating semantic identifiers (SIDs) in generative retrieval and recommendation, combining a dataset release with offline and limited online analyses. Reviewers (and AC) agree that the dataset is large and timely, and a useful contribution for the community, and that the empirical study offers practical insights into SID design choices (ZTNg, kG4V, 4riH).  However, there are several concerns which prevent an accept recommendation at this time:

- many technical components are repurposed from prior work, leaving the core novelty concentrated mainly in benchmarking and post-hoc analysis rather than new methods (JeAR, kG4V); this is OK, but despite revisions, the paper’s organization and positioning relative to strong baselines, and clarity around evaluation and deployment assumptions are weaker than expected for a leading benchmark (ZTNg, JeAR)

- the release is limited to frozen embeddings rather than raw text and images, significantly constraining the benchmark’s flexibility and long-term research value (JeAR, 4riH); the derivation of SIDs is half the reproducibility element for GR techniques in the modern era.

- although several reviewers raised their scores after rebuttal, these issues collectively limit the paper’s impact and generality, and I therefore lean toward rejection, encouraging resubmission after a clearer scoping of contributions, and scoping as a benchmark paper with more holistic analyses/comparisons (to other datasets in the community, unique features of this dataset, unique conclusions drawn from this dataset which are supported/disagree with others, and comparison of multiple baseline methods besides just GR ones).  Authors did some of these in rebuttal but it doesn't fix the overarching positioning problems in the paper which should robustify this positioning and emphasize the appropriate contributions.

- finally, the template seems to be using a different font than ICLR style guide which poses a potential fairness risk compared to other papers who meet these guidelines.

**Reviewer Concerns:**

See above.

**Reviewer Scores:**

JeAR: 4->5
ZTNg: 4->6
kG4V: 4->5
4riH: 6->probably 7

---

### Decision · Program_Chairs · 2026-01-26

Reject